# Enhancing Spectral GNNs: From Topology and Perturbation Perspectives

Taoyang Qin [1]    Ke-Jia CHEN [1 2]    Zheng Liu [1]

## Abstract

Spectral Graph Neural Networks process graph signals using the spectral properties of the normalized graph Laplacian matrix. However, the frequent occurrence of repeated eigenvalues limits the expressiveness of spectral GNNs. To address this, we propose a higher-dimensional sheaf Laplacian matrix, which not only encodes the graph's topological information but also increases the upper bound on the number of distinct eigenvalues. The sheaf Laplacian matrix is derived from carefully designed perturbations of the block form of the normalized graph Laplacian, yielding a perturbed sheaf Laplacian (PSL) matrix with more distinct eigenvalues. We provide a theoretical analysis of the expressiveness of spectral GNNs equipped with the PSL and establish perturbation bounds for the eigenvalues. Extensive experiments on benchmark datasets for node classification demonstrate that incorporating the perturbed sheaf Laplacian enhances the performance of spectral GNNs.

## 1. Introduction

Both spatial and spectral Graph Neural Networks (GNNs) have shown outstanding performance in various graph tasks (Kipf & Welling, 2017; Gilmer et al., 2017; Huang et al., 2023; Jin et al., 2024). Unlike spatial GNNs, spectral GNNs conduct frequency-domain convolution on normalized Laplacian matrices of graphs. However, in most graph datasets, the multiplicity of eigenvalues of the normalized graph Laplacian matrices is not small. The repeated eigenvalues limit the expressive power of spectral GNNs, resulting in a deteriorated performance (Wang & Zhang, 2022; Lu et al., 2024).

[1]School of Computer Science, Nanjing University of Posts and Telecommunications, Nanjing, China [2]Jiangsu Key Laboratory of Big Data Security & Intelligent Processing, Nanjing, China. Correspondence to: Ke-Jia CHEN <chenkj@njupt.edu.cn>, Zheng Liu <zliu@njupt.edu.cn>.

*Proceedings of the 42nd International Conference on Machine Learning*, Vancouver, Canada. PMLR 267, 2025. Copyright 2025 by the author(s).

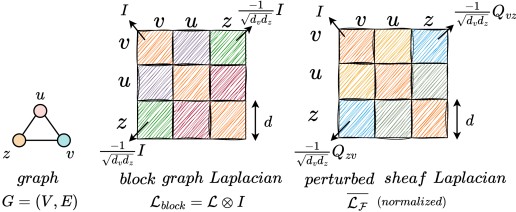

*Figure 1.* The perturbed sheaf Laplacian $\overline{\mathcal{L}_\mathcal{F}}$ can be viewed as a special perturbation of $\mathcal{L}_{block}$ ($\mathcal{L}$ is the normalized graph Laplacian, $\otimes$ denotes the Kronecker product, $I$ is the $d \times d$ identity matrix, and all restriction maps are orthogonal). Specifically, for each $(v, z) \in E$, if $Q_{vz}$ slightly deviates from the identity $I$ while satisfying the sheaf Laplacian constraints, the resulting matrix is both a valid normalized sheaf Laplacian and a perturbed version of $\mathcal{L}_{block}$.

Wang and Zhang (Wang & Zhang, 2022) highlight the issue of repeated eigenvalues and propose an orthogonal Jacobi basis approach in which each prediction dimension has a flexible filter design. However, the individual filter design does not fully resolve the performance degeneracy caused by different eigenvectors sharing the same eigenvalue. Lu et al. (Lu et al., 2024) tackle the issue by adding a diagonal matrix of uniformly sampled eigenvalues to the normalized Laplacian matrix. The polynomial filters integrated with the corrected eigenvalues yield improved node classification results. Although this method helps mitigate repeated eigenvalues, it also compromises the original topological information encoded in the normalized Laplacian matrix by reducing the operator's spectral gap, which reflects graph's connectivity and information flow (see Appendix F.2).

Besides the above issues, both works focus only on linear settings, leaving the impact of nonlinear activation function unexplored.

We focus on the challenge of repeated eigenvalues in a nonlinear context. We propose to utilize a higher dimensional sheaf Laplacian matrix, which guarantees a higher upper bound on the number of distinct eigenvalues (Hansen & Ghrist, 2018; Hansen, 2020). Specifically, we normalize the sheaf Laplacian matrix by applying perturbation on the block form of the normalized graph Laplacian matrix. The designed perturbation of the identity matrix serves as the restriction map, which increases the diversity of eigenvalue

while retaining the topological information of the normalized Laplacian matrix (as shown in Figure 1). To summarize, our contributions are as follows:

- We propose a perturbed sheaf Laplacian (PSL) matrix which has more distinct eigenvalues by carefully designing the perturbation on the basis of perturbation theory.

- We achieve optimal model performance by applying the PSL matrix on popular spectral GNNs architectures and optimizing the perturbation process.

- We theoretically analyze the expressiveness of the PSL-based GNNs, and characterize the impact of the perturbations by establishing the perturbation bounds for the eigenvalues.

Experiments on benchmark datasets demonstrate the superiority of PSL-based spectral GNNs. The increased number of distinct eigenvalues of the learned perturbed sheaf Laplacian matrices validates the effectiveness of the proposed perturbation method.

## 2. Preliminaries

This section provides a brief introduction to spectral GNNs and highlights the key aspects of the theory of cellular sheaves. All important notations are given in Appendix A.

### 2.1. Spectral Graph Neural Networks

Spectral GNNs build on spectral graph theory (Chung, 1997), which applies the graph Fourier transform to define frequency-domain convolutions. For a graph $G = (V, E)$ with $n$ nodes, the normalized Laplacian is $\mathcal{L} = I - D^{-\frac{1}{2}} A D^{-\frac{1}{2}}$. Its eigendecomposition is $\mathcal{L} = U \Lambda U^T$, where $\Lambda$ is diagonal and $U$ holds the eigenvectors. The graph Fourier transform of a signal $X \in \mathbb{R}^{n \cdot f}$ is $\hat{X} = U^T X$, with each row of $\hat{X}$ representing a distinct frequency component. The output $Z$ of a spectral convolution can be expressed as:

$$Z = \mathbf{g} *_G X = U\left((U^T \mathbf{g}) \odot (U^T X)\right) = U\hat{G}U^T X \quad (1)$$

where $\mathbf{g}$ is the spatial kernel, and $\hat{G} = \mathrm{diag}(\hat{\mathbf{g}}_1, \ldots, \hat{\mathbf{g}}_n)$ denotes the spectral kernel coefficients. For spectral GNNs, a polynomial function is commonly used to approximate different kernels, which can be expressed as follows:

$$\begin{aligned} Z &= \varphi(U g(\Lambda) U^T X W) \\ &= \varphi(U \left(\sum_{k=0}^{K} \alpha_k \Lambda^k\right) U^T X W) \\ &= \varphi(\sum_{k=0}^{K} \alpha_k \mathcal{L}^k X W), \end{aligned} \quad (2)$$

where $g$ is the filter function, $W$ is a learnable weight matrix, $\alpha_k$ is a trainable coefficient for the $k$-th order polynomial approximation, and $\varphi$ is a function like multi-layer perceptrons (MLPs).

Note that, if $\varphi$ is a linear function, spectral GNNs are termed linear spectral GNNs; if $\varphi$ is a nonlinear function, they are termed nonlinear spectral GNNs.

### 2.2. Theory of Cellular Sheaves

The theory of cellular sheaves, rooted in algebraic topology, has been incorporated into GNNs to additionally augment the graph's topological structure and thereby improve the performance of GNNs (Bodnar et al., 2022; Duta et al., 2023a).

**Definition 2.1.** A cellular sheaf $(G, \mathcal{F})$ over an undirected graph $G = (V, E)$ consists of:

- A vector space $\mathcal{F}(v)$ associated with each node $v \in V$.

- A vector space $\mathcal{F}(e)$ associated with each edge $e \in E$.

- A linear map $\mathcal{F}_{v \trianglelefteq e}$ for each incident node-edge pair $v \trianglelefteq e$, mapping vectors from $\mathcal{F}(v)$ to $\mathcal{F}(e)$.

Here, the vector spaces for nodes and edges are termed stalks, and the linear maps between them are termed restriction maps. We set all the stalks to be $\mathbb{R}^d$ and confine all restriction maps to orthogonal mappings. The restriction map $\mathcal{F}_{v \trianglelefteq e} : \mathcal{F}(v) \to \mathcal{F}(e)$ maps vectors from node stalk to edge stalk and $\mathcal{F}_{v \trianglelefteq e}^\top : \mathcal{F}(e) \to \mathcal{F}(v)$ maps vectors from edge stalk to node stalk. It follows that $\mathcal{F}_{u \trianglelefteq (v,u)}^\top \mathcal{F}_{v \trianglelefteq (v,u)}$ maps vectors from $\mathcal{F}(v)$ to $\mathcal{F}(u)$, where $(v, u) \in E$.

**Definition 2.2.** The sheaf Laplacian matrix of a sheaf over an undirected graph $G = (V, E)$ is a block matrix $\mathcal{L}_\mathcal{F}$. The diagonal blocks are $\mathcal{L}_{\mathcal{F},vv} = \sum_{v \trianglelefteq e} \mathcal{F}_{v \trianglelefteq e}^\top \mathcal{F}_{v \trianglelefteq e}$, while the non-diagonal blocks are $\mathcal{L}_{\mathcal{F},vu} = -\mathcal{F}_{v \trianglelefteq e}^\top \mathcal{F}_{u \trianglelefteq e}$, where $(v, u) \in E$.

The sheaf Laplacian matrix is positive semidefinite, which means that the eigenvalues are non-negative (Hansen, 2020; Bodnar et al., 2022). Additionally, when all the restriction maps are identity maps, the sheaf Laplacian matrix degenerates into the block form of the graph Laplacian matrix.

**Definition 2.3.** The normalized form of the sheaf Laplacian matrix is given by $\overline{\mathcal{L}_\mathcal{F}} = D^{-\frac{1}{2}} \mathcal{L}_\mathcal{F} D^{-\frac{1}{2}}$, where $D_v = \mathcal{L}_{\mathcal{F},vv}$ and the diagonal matrix of the sheaf Laplacian matrix is $D = \mathrm{diag}(D_1, \ldots, D_n)$.

In the following, we will refer to the normalized sheaf Laplacian matrix simply as the sheaf Laplacian.

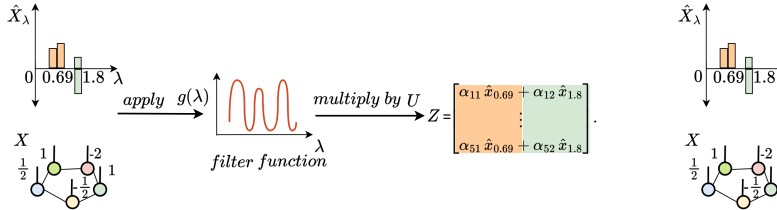

(a) Limited predictive ability.

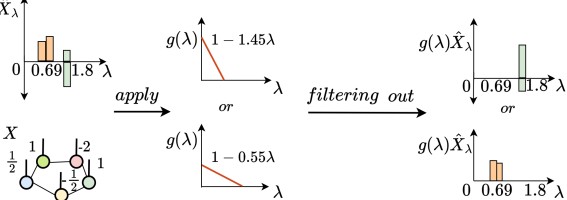

(b) Loss of frequency components.

*Figure 2.* $X$ is the graph data, and $\hat{X}_\lambda$ its Fourier-transformed matrix. $g$ is the filter function. (a) Because the graph has only two distinct nonzero eigenvalues, each element in Z is a linear combination of just two types of frequency components, thereby limiting the expressive power of spectral GNNs. (b) Poorly designed filter functions can lead to the loss of frequency components, thereby limiting the expressive power of spectral GNNs.

## 3. Motivation

In this section, we summarize and extend the discussion of repeated eigenvalue issues as presented by Wang and Zhang (Wang & Zhang, 2022) and Lu et al. (Lu et al., 2024), which contribute to the motivation for our work.

### 3.1. Limited Predictive Ability

The phenomenon of repeated eigenvalues limits the predictive ability of spectral GNNs, thus affecting their expressive power (as shown in Figure 2 (a)). Firstly, we use a lemma to describe this issue.

**Lemma 3.1.** *When there are only $k$ distinct eigenvalues of the normalized Laplacian matrix, linear spectral GNNs can produce at most $k$ different filter coefficients, and thus can only generate one-dimensional predictions with a maximum of $k$ arbitrary elements.*

The proof of this lemma can be found in Appendix A.1 of Lu et al. (Lu et al., 2024). This lemma reveals a key problem: the predictive ability of linear Spectral GNNs is limited by the number of distinct eigenvalues of the normalized graph Laplacian matrix.

In this paper, we extend the lemma to the case of nonlinear spectral GNNs which use an activation function (Rosenblatt, 1958; Nair & Hinton, 2010; Goodfellow et al., 2016).

**Theorem 3.2.** *When there are only $k$ distinct eigenvalues of the normalized Laplacian matrix, and the nonlinear real-valued function $\sigma$ satisfies $\sigma(x) \neq 0$ for $x \neq 0$, nonlinear spectral GNNs can produce at most $k$ different filter coefficients, and thus can only generate one-dimensional predictions with a maximum of $k$ arbitrary elements.*

Theorem 3.2 shows that, even with real-valued activation functions such as tanh and sigmoid, the predictive ability of nonlinear spectral GNNs remains equivalent to that of linear spectral GNNs, both of which are similarly limited. This limitation arises from the issue of repeated eigenval-

ues of the normalized graph Laplacian matrix. The proof is in Appendix B.

### 3.2. Loss of Frequency Components

Another issue caused by repeated eigenvalues is that if the filter function $g(\lambda)$ is poorly designed, setting a specific eigenvalue to zero can lead to the disappearance of corresponding frequency components (As shown in Figure 2 (b)).

Interestingly, for many commonly used graph datasets, the eigenvalue 1 of the corresponding normalized graph Laplacian matrix has the highest multiplicity (see Appendix F.5 for details). This observation may also provide a new perspective on why adding self-loops to the adjacency matrix leads to better GCN performance on certain datasets (Kipf & Welling, 2017; Wu et al., 2019). Given that the filter function of GCN is $1 - \lambda$, it filters out all frequency components associated with the eigenvalue 1. By introducing self-loops, the distribution of eigenvalues is altered, thereby causing GCN to lose fewer frequency components.

### 3.3. Restricted Frequency Processing Capability

The phenomenon of repeated eigenvalues also restricts the frequency processing capability of spectral GNNs. Consider $\hat{X} = U^T X W$, where $g(\Lambda)$ represents the filtering function, with $\Lambda$ being the eigenvalue matrix of the normalized Laplacian. Evidently, $g(\Lambda)\hat{X}$ is equivalent to $g(\lambda_i) \odot \hat{X}_i$, where $g(\lambda_i)$ scales each $\hat{X}_i$ element-by-element. Thus, for identical eigenvalues $\lambda_i = \lambda_j$, the associated frequency components $\hat{X}_i$ and $\hat{X}_j$ are scaled by the same factor, which limits the expressiveness of the spectral GNNs.

### 3.4. Why Perturbed Sheaf Laplacian

The solution to the above issues is to construct a Laplacian matrix with more distinct eigenvalues. The sheaf Lapla-

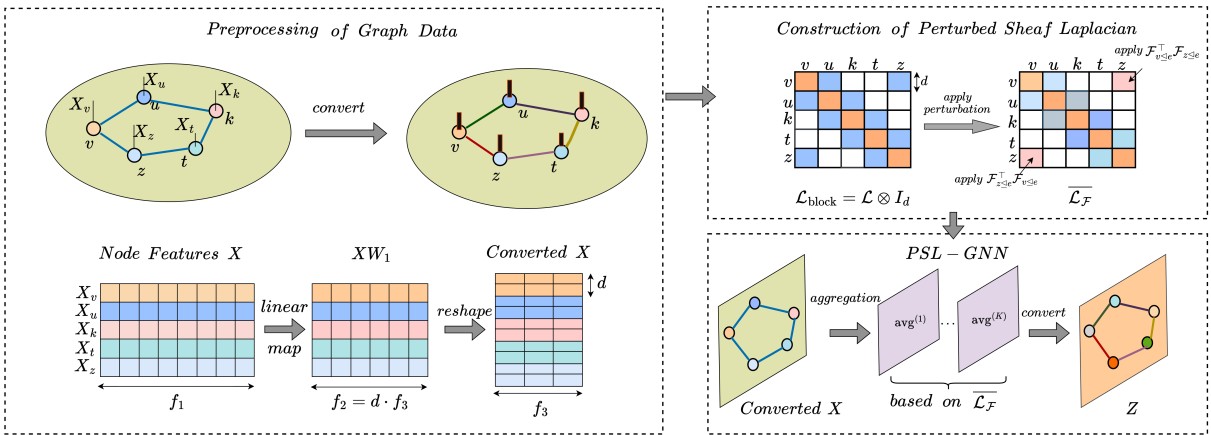

Figure 3. Model workflow: The convert operation initially reduces feature dimensionality and reshapes the input to $X \in \mathbb{R}^{nd \times k}$. The $l$-th aggregation step, $\text{avg}^{(l)}$, is performed as $\text{Combine}(\overline{\mathcal{L_F}}^{(l)}, X^{(l)}, W^{(l)})$. After final aggregation, and the convert operation means reshaping the output back to $n$ rows of $d \times k$ columns, followed by a linear transformation to yield $Z$ with columns equal to the number of node classes.

cian is from the theory of cellular sheaves. When applied to graphs, the sheaf Laplacian naturally provides a larger set of eigenvalues while maintaining a positive semidefinite nature and the eigenvalue range in $[0, 2]$ (Hansen, 2020).

We carefully design the restriction map as a slight perturbation of the identity matrix in order to construct a sheaf Laplacian with more distinct eigenvalues. We call it the **perturbed sheaf Laplacian** (**PSL**). The perturbed sheaf Laplacian matrix achieves two advantages: (1) By aligning with perturbation theory, the PSL matrix increases the eigenvalue diversity; (2) the PSL matrix is structurally close to the normalized graph Laplacian matrix, retaining its topological information.

## 4. Method

This section introduces the proposed PSL-GNN model (with its general workflow illustrated in Figure 3). The complexity of our method is illustrated in Appendix F.1.

### 4.1. Preprocessing of Graph Data

To make the first dimension of the input matrix $X_0 \in \mathbb{R}^{n \times f_1}$ compatible with the second dimension of the sheaf Laplacian matrix $\overline{\mathcal{L_F}} \in \mathbb{R}^{nd \times nd}$, we first apply a linear projection:

$$X_0 W_1 \in \mathbb{R}^{n \times d f_3}, \qquad (3)$$

where $W_1 \in \mathbb{R}^{f_1 \times f_2}$ with $f_2 = d \times f_3$. Next, we reshape the resulting matrix:

$$X = \mathcal{R}\big(X_0 W_1, \ \mathbb{R}^{nd \times f_3}\big), \qquad (4)$$

where $\mathcal{R}(X_0 W_1, \mathbb{R}^{nd \times f_3})$ represents reshaping $X_0 W_1$ into a matrix with $nd$ rows and $f_3$ columns.

### 4.2. Construction of Perturbed Sheaf Laplacian

When there exists a restriction map that is not an identity map, $\overline{\mathcal{L_F}}$ can be viewed as the result of a special perturbation applied to the block form of the normalized graph Laplacian matrix $\mathcal{L}_{\text{block}}$. For each block element $\mathcal{L}_{\text{block},ij}$, the corresponding perturbation matrix $Q_{ij} = \mathcal{F}_{i \trianglelefteq e}^{\top} \mathcal{F}_{j \trianglelefteq e}$ is applied, resulting in $\overline{\mathcal{L_F}}$. According to Section 3.4, when $\mathcal{F}_{i \trianglelefteq e}^{\top} \mathcal{F}_{j \trianglelefteq e}$ slightly deviates from the identity matrix, the corresponding sheaf Laplacian matrix is regarded as a perturbed sheaf Laplacian matrix.

Based on this, $Q_{ij}$ can be viewed as a perturbation applied to $L_{\text{block},ij}$. To ensure that the overall perturbation effect increases the number of distinct eigenvalues of $\overline{\mathcal{L_F}}$, we propose the following theorem to guide the perturbations.

**Theorem 4.1.** *Let $P = \overline{\mathcal{L_F}} - \mathcal{L}_{block}$ be the perturbation matrix applied to $\mathcal{L}_{block}$, and let $\phi = \min_{i,j;\, i \neq j} |\lambda_i - \lambda_j|$, where $\lambda_i$ are the eigenvalues of $\mathcal{L}_{block}$. If $\|P\|_2 < \frac{\phi}{2}$, then the multiplicity of eigenvalues of $\mathcal{L_F}$ will decrease.*

The proof is in Appendix B. According to Theorem 4.1, to reduce the multiplicity of eigenvalues, the orthogonal matrices of different blocks should be restricted to slightly deviate from the identity matrices. Considering that the reflection matrix can deviate from the identity matrix to varying extents by adjusting the reflection vector, we constrain every restriction map to the reflection matrix. Specifically, we assign every restriction map to $\mathcal{F}_{i \trianglelefteq (i,j)} = I - 2\frac{\mathbf{u}_{ij} \mathbf{u}_{ij}^{\top}}{\|\mathbf{u}_{ij}\|^2}$, where $\mathbf{u}_{ij}$ is a vector of dimensions $d$ ($d$ represents the dimension of the stalks). Here, $\mathbf{u}_{ij}$ is parameterized and learnable, allowing $Q_{ij}$ to deviate from the identity matrix. To ensure that the applied perturbations remain minimal, we constrain each component of $\mathbf{u}_{ij}$ within $[\frac{\eta}{10}, \eta]$ for the reflection matrix, where $\eta$ is a sufficiently small value.

### 4.3. PSL-GNN: Perturbed Sheaf Laplacian-Based Graph Neural Network

The proposed perturbed sheaf Laplacian matrix can be integrated with any spectral GNNs. In this section, we apply it within the GCN (Kipf & Welling, 2017) and GPRGNN (Chien et al., 2021) respectively as examples.

#### 4.3.1. PSL-GCN

The architecture of GCN is often represented as:

$$X_0^{(l+1)} = \sigma((I - \Delta)X_0^{(l)}W), \tag{5}$$

where $\Delta$ represents the normalized graph Laplacian matrix, $l$ denotes the layer index, and $X^{(l)}$ represents the feature matrix at layer $l$. Through integration, we obtain PSL-GCN, which can be expressed as follows:

$$X^{(l+1)} = \sigma((I - \Delta_{\mathcal{F}})X^{(l)}W_2), \tag{6}$$

where $X^{(l+1)} \in \mathbb{R}^{nd \times f_4}$ and $X^{(l)} \in \mathbb{R}^{nd \times f_3}$. $\Delta_{\mathcal{F}} \in \mathbb{R}^{nd \times nd}$ is the PSL matrix, and $W_2 \in \mathbb{R}^{f_3 \times f_4}$ is the parameter matrix used for linear transformations within the network.

We need to reshape the output of the last layer and perform a series of transformations such as $W_{last} \in \mathbb{R}^{df_{last} \times c}$ and $\mathcal{R}$ to obtain the final prediction matrix $Z$:

$$Z = \mathcal{R}(X^{(last)}, \mathbb{R}^{n \times df_{last}})W_{last}. \tag{7}$$

#### 4.3.2. PSL-GPR

The structure of GPRGNN is represented as follows:

$$Z = \sum_{k=0}^{K} \alpha_k (I - \mathcal{L})^k X_0, \tag{8}$$

where $X_0 \in \mathbb{R}^{n \times f_1}$ represents the feature matrix, and $\mathcal{L}$ represents the normalized graph Laplacian matrix. According to Equation (4), we have the converted feature matrix $X \in \mathbb{R}^{nd \times f_3}$. Following the architecture of GPRGNN, our model, namely PSL-GPR, is expressed as:

$$Y = \sum_{k=0}^{K} \alpha_k (I - \overline{\mathcal{L}_{\mathcal{F}}})^k X. \tag{9}$$

In this equation, $Y \in \mathbb{R}^{nd \times f_3}$ represents the output feature matrix, $I \in \mathbb{R}^{nd \times nd}$ is the identity matrix, $\overline{\mathcal{L}_{\mathcal{F}}} \in \mathbb{R}^{nd \times nd}$ is the PSL matrix, $K$ is the order of the polynomial, and $\alpha_k$ are the trainable coefficients.

We reshape $Y$ into $\mathbb{R}^{n \times df_3}$ and apply a linear transformation with $W_3 \in \mathbb{R}^{df_3 \times c}$ to obtain the final prediction matrix $Z \in \mathbb{R}^{n \times c}$, where $c$ represents the number of classes. This process can be expressed as:

$$Z = \mathcal{R}(Y, \mathbb{R}^{n \times df_3})W_3. \tag{10}$$

## 5. Theoretical Analysis

In this section, we conduct a theoretical analysis of the expressive power of PSL-GNN and establish the perturbation bounds of the eigenvalues. All proofs are in Appendix C and Appendix D.

### 5.1. Expressiveness of PSL-GNN

We first discuss the expressive power of the block matrix form $\mathcal{L}_{block}$ of the normalized graph Laplacian matrix $\mathcal{L}$.

**Proposition 5.1.** *The number of distinct eigenvalues of $\mathcal{L}_{block} \in \mathbb{R}^{nd \times nd}$ is the same as that of $\mathcal{L} \in \mathbb{R}^{n \times n}$, where $\mathcal{L}$ is the normalized graph Laplacian matrix and $\mathcal{L}_{block} = \mathcal{L} \otimes I$, with $I \in \mathbb{R}^{d \times d}$ being the identity matrix. Also, their corresponding GNNs, which share the same architecture, are mutually equivalent.*

Proposition 5.1 indicates that when the applied orthogonal matrices are the identity matrices, PSL-GNN is equivalent to the corresponding GNN.

**Corollary 5.2.** *PSL-GNN has stronger one-dimensional prediction ability.*

For a given normalized graph Laplacian matrix, its PSL possesses a greater number of distinct eigenvalues, which underpins its enhanced predictive ability.

**Corollary 5.3.** *PSL-GNN loses fewer frequency components.*

With fewer repeated eigenvalues than the normalized Laplacian, PSL-GNN remains robust even if high-multiplicity frequency components are discarded.

**Corollary 5.4.** *PSL-GNN has a stronger capability in processing frequency components.*

The PSL possesses a greater number of distinct eigenvalues, allowing a more diverse treatment of frequency components.

### 5.2. Perturbation Bounds of Eigenvalues

We establish the perturbation bounds for the eigenvalues based on perturbation theory (Weyl, 1912; Dailey et al., 2014).

**Theorem 5.5.** *Let $\mathcal{L}_{block}$ and $\mathcal{L}_{\mathcal{F}} = \mathcal{L}_{block} + P$ be $N \times N$ symmetric positive semidefinite matrices, with eigenvalues $\lambda_1, \ldots, \lambda_N$ and $\tilde{\lambda}_1, \ldots, \tilde{\lambda}_N$, respectively, where $\mathcal{L}_{block}$ is diagonally dominant. Suppose the perturbation matrix $P$ satisfies $|P_{ij}| \leq \epsilon |(L_{block})_{ij}|$ for all $i \neq j$ for all $i$, where $0 < \epsilon < \frac{1}{2}$. Hence, for each $i$, $|\tilde{\lambda}_i - \lambda_i| \leq \frac{2\epsilon}{1-2\epsilon} \tilde{\lambda}_{max}$.*

As the size of the perturbation $\eta$ decreases, the corresponding $\epsilon$ also decreases, resulting in a smaller range of eigenvalue perturbations.

# 6. Experiment

In this section, we give the results of the performance comparison between PSL-GNN and GNN baselines on the node classification task. Besides, we validate the effectiveness of the proposed perturbed sheaf Laplacian matrix through ablation studies. We also perform a statistical analysis of the independent eigenvalues of the final learned perturbed sheaf Laplacian matrices. The experimental setup is in Appendix E. Additional experimental studies can be found in Appendix F.

## 6.1. Evaluation on Real-World Datasets

We conduct experiments on 7 benchmark graph datasets (all information about the datasets is summarized in Appendix E.2.), and the results are presented in Table 1. We evaluate the accuracy of the PSL-GNNs by comparing them with their corresponding GNN models. We also calculate the vertical average gain (VAG) of the perturbed sheaf Laplacian matrix for all models on each dataset. The formula for VAG is given as follows:

$$VAG = \frac{1}{|\mathcal{M}|} \sum_{m_i \in \mathcal{M}} \left( \text{ACC}(m_i(\mathcal{F})) - \text{ACC}(m_i(G)) \right),$$

$$(11)$$

where $\mathcal{M}$ is the set of GNN models, ACC is the short form of accuracy, $G$ is the original graph, and $\mathcal{F}$ is a sheaf over the graph. We also calculate the horizontal average gain (HAG) of the perturbed sheaf Laplacian matrix for all datasets on each model. The formula for HAG is given as follows:

$$HAG = \frac{1}{|\mathcal{D}|} \sum_{d_i \in \mathcal{D}} \left( \text{ACC}(d_i(\mathcal{F})) - \text{ACC}(d_i(G)) \right), \quad (12)$$

where $\mathcal{D}$ is the set of used datasets.

By analyzing the results in Table 1, we have the following observations: (1) All models are enhanced by the perturbed sheaf Laplacian matrices. This improvement can be attributed to the fact that the perturbed sheaf Laplacian matrices have more distinct eigenvalues. Consequently, the PSL-GNNs have a stronger capability to handle various frequency components, thereby improving their performance in the node classification task; (2) For Texas and Cornell, which are characterized by a relatively small number of nodes, the performance improvement is particularly significant. The observed phenomenon can be attributed to the inherently lower structural stability of smaller graphs, which makes them more sensitive to perturbations. This heightened sensitivity facilitates the splitting of eigenvalues into distinct values, thereby significantly increasing the number of distinct eigenvalues; (3) The GCN shows the highest average gain compared to other baselines. This can be explained from the perspective of suppressing the loss

of frequency components: Unlike the variable filter functions in other baselines, the GCN filter function is fixed, i.e., $g(\lambda) = 1 - \lambda$, where $\lambda$ represents the eigenvalue of the normalized graph Laplacian matrix. The multiplicity of $\lambda = 1$ is relatively high (see Figure 5 for details), which leads to a significant loss of frequency components corresponding to $\lambda = 1$, thereby severely affecting the expressive power of spectral GNNs. In contrast, the perturbed sheaf Laplacian can reduce the multiplicity of $\lambda = 1$, resulting in a higher average gain.

## 6.2. Ablation Study

To further study the effectiveness of the perturbed sheaf Laplacian matrix, we compare it with both the general sheaf Laplacian matrix (GSL) (Bodnar et al., 2022) and the normalized graph Laplacian matrix.

The construction of the GSL involves using $\frac{d(d-1)}{2}$ parameters to generate each restriction map $\mathcal{F}_{v \trianglelefteq e}$. Besides, each parameter is no longer constrained by $\eta$. Based on the above setup, the restriction maps are allowed to fully explore the orthogonal group. For the construction of normalized graph Laplacian matrix, we set $\eta = 0$, which means no perturbation is applied. Consequently, the perturbed sheaf Laplacian matrix degenerates into the block matrix form of the normalized graph Laplacian matrix. We integrate these three types of Laplacian into vanilla GNN, creating PSL-GNN, GSL-GNN, GNN (vanilla). Subsequently, we compare their performance on node classification. All models share the same parameters, except $\eta$, which is tuned for PSL-GPR and PSL-GCN ($\{1e-1, 1e-2, 1e-3, 1e-4\}$) and set to 0 for GPR-GNN and GCN. This ensures fair and reliable comparisons.

We conduct multiple training runs and averaged various metrics to obtain the reliable results in Table 2. The results demonstrate that: (1) GSL-GNN outperforms vanilla GNN, probably because the general sheaf Laplacian generalizes the graph Laplacian, offering greater learnability and flexibility for downstream tasks; (2) The performance of PSL-GNN is comparable to that of GSL-GNN. It indicates that the structure learned by the general sheaf Laplacian might align closely with that learned by the perturbed sheaf Laplacian.

## 6.3. The Number of Distinct Eigenvalues

We measure the number of distinct eigenvalues for both the normalized graph Laplacian matrix and the final learned perturbed sheaf Laplacian matrix when $\eta \geq 1e-3$, respectively. In Table 3, $N_G$ denotes the number of distinct eigenvalues of the normalized graph Laplacian matrix, while the other entries represent the number of distinct eigenvalues of the learned perturbed sheaf Laplacian matrix. As the

*Table 1.* Experimental results on node classification task: mean accuracy (%) ±95% confidence interval.

| Datasets | Cora | Pubmed | Citeseer | Photo | Cornell | Texas | Actor | **HAG** |
|---|---|---|---|---|---|---|---|---|
| GCN | 86.62±1.06 | 86.51±0.47 | 75.32±1.91 | 87.52±0.58 | 75.32±1.98 | 78.12±1.87 | 34.23±1.54 | |
| PSL-GCN | **88.09**±1.75 | **88.21**±2.43 | **76.75**±2.32 | **90.12**±0.90 | **77.89**±2.45 | **80.45**±2.21 | **35.67**±1.83 | 1.93 ↑ |
| APPNP | 88.11±1.35 | 87.39±0.88 | 76.34±1.71 | 90.23±0.50 | 78.56±2.01 | 80.12±1.83 | 34.89±1.73 | |
| PSL-APPNP | **89.03**±1.79 | **88.23**±2.39 | **76.92**±2.36 | **90.58**±0.94 | **79.89**±2.37 | **82.34**±2.19 | **36.01**±2.11 | 1.05 ↑ |
| Graph-Heat | 86.85±1.29 | 87.12±1.69 | 74.75±2.44 | 89.32±0.74 | 75.78±1.88 | 79.45±1.72 | 34.01±1.62 | |
| PSL-Heat | **87.38**±1.75 | **88.74**±2.42 | **76.72**±2.52 | **89.43**±0.93 | **77.34**±2.29 | **80.67**±2.14 | **35.56**±1.94 | 1.22 ↑ |
| BernNet | 88.29±0.63 | 88.50±1.19 | 76.78±0.49 | 94.02±1.26 | 86.23±2.14 | 85.67±1.99 | 36.12±1.49 | |
| PSL-BernNet | **89.67**±1.79 | **89.54**±2.40 | **77.46**±2.38 | **94.19**±0.94 | **88.34**±2.48 | **86.89**±2.32 | **37.45**±1.91 | 1.13 ↑ |
| GPRGNN | 87.59±1.31 | 88.01±1.14 | 76.83±1.67 | 92.98±1.09 | 86.32±2.11 | 85.17±1.87 | 36.42±1.65 | |
| PSL-GPR | **88.45**±1.48 | **89.11**±1.89 | **77.42**±2.02 | **94.23**±1.47 | **88.67**±2.33 | **86.45**±2.04 | **37.45**±1.92 | 1.21 ↑ |
| Jacobi | 88.83±1.65 | 88.03±0.75 | 76.89±1.84 | 93.72±1.53 | 87.11±2.07 | 84.45±1.96 | 36.56±1.78 | |
| PSL-Jacobi | **89.01**±1.80 | **89.02**±2.42 | **77.71**±2.50 | **94.43**±0.96 | **88.98**±2.42 | **86.23**±2.15 | **37.78**±2.05 | 1.08 ↑ |
| **VAG** | 0.89 ↑ | 1.22 ↑ | 1.01 ↑ | 0.86 ↑ | 1.96 ↑ | 1.68 ↑ | 1.28 ↑ | |

table shows, the number of distinct eigenvalues of the perturbed sheaf Laplacian matrix significantly exceeds that of the normalized graph Laplacian matrix.

# 7. Related work

In this section, we outline the research developments related to the issues of interest, along with the background and tools employed.

## 7.1. Expressiveness of GNNs

The research on expressive power of GNNs originates from the Weisfeiler-Lehman (WL) test for graph isomorphism (Weisfeiler & Leman, 1968). Building on this foundation, GIN (Xu et al., 2019b) establishes a direct connection between the expressive power of GNNs and the 1-WL test. Since then, various works have attempted to analyze GNNs with the WL test and graph isomorphism testing (Morris et al., 2019; Maron et al., 2019a; Chen et al., 2019; Zhang et al., 2021). Besides, some studies measure the expressive power of GNNs by using alternative approaches, such as expressing universal invariant functions (Maron et al., 2019b) and counting graph's substructures (Chen et al., 2020). Balcilar et al.(Balcilar et al., 2021) first analyze the expressive power of GNNs from a spectral perspective. Wang and Zhang (Wang & Zhang, 2022) show that the number of distinct eigenvalues in the normalized graph Laplacian matrix impacts the expressive power of linear spectral GNNs. Besides, they also build a bridge between the expressive power of spectral GNNs and 1-WL test. Lu et al. (Lu et al., 2024) further analyze that the presence of repeated eigenvalues imposes limitations on the predictive ability of linear spectral GNNs. In this paper, we summarize and extend the work of Wang and Zhang, as well as Lu et al., to address the limitations posed by repeated eigen-

values of normalized graph Laplacian matrix.

## 7.2. Sheaf Neural Networks on Graphs

Sheaf Neural Networks (SNNs), introduced by Hansen and Gebhart (Hansen & Gebhart, 2020), extend graph neural networks by using the sheaf Laplacian matrix for dealing with non-constant, heterogeneous, and signed node relationships. Building on this foundation, Bodnar et al. (Bodnar et al., 2022) apply sheaf Laplacian matrix to address heterophily and oversmoothing problems in graph neural networks. This is achieved by concatenating node features as parameters, which are then used to construct restriction maps. Barbero et al. (Barbero et al., 2022) generate a sheaf Laplacian matrix through a pre-trained approach, which is suitable for manifold data and effectively incorporates the underlying geometric characteristics of the data. Duta et al. (Duta et al., 2023b) construct the hyper sheaf Laplacian to capture both linear and non-linear diffusion by selecting the most discrepant feature pairs within hyperedges. Battiloro et al. (Battiloro et al., 2024) use VDM (Singer & Wu, 2012) to construct the sheaf Laplacian and design a tangent bundle filter. This method requires a large amount of specialized data and involves high complexity. Our method simplifies the process by perturbing the block form of the normalized graph Laplacian, which has been proven both efficient and effective.

## 7.3. Eigenvalue Perturbation Theory

Early research mainly focuses on eigenvalue perturbation of general symmetric matrices. Tools such as Weyl's theorem (Weyl, 1912) and Gershgorin's disk theorem (Gerschgorin, 1931) are applied to derive the fundamental effects of perturbations on eigenvalues. Due to the stability of semidefinite matrices and diagonally dominant matrices,

*Table 2.* Comparison of models based on three Laplacian matrices (GNN vs. GSL-GNN vs. PSL-GNN): mean accuracy (%) ±95 (%) on seven datasets.

| Datasets | Cora | Pubmed | Citeseer | Photo | Texas | Cornell | Actor |
|---|---|---|---|---|---|---|---|
| GCN | 86.8±1.1 | 86.4±0.8 | 75.4±2.2 | 87.6±1.5 | 78.4±1.9 | 75.2±1.7 | 34.1±1.5 |
| GSL-GCN | **87.6**±0.9 | **86.9**±2.0 | **76.4**±2.1 | **89.4**±1.1 | **80.1**±1.3 | **77.7**±1.3 | **35.3**±1.9 |
| PSL-GCN | **88.1**±0.8 | **87.9**±1.8 | **77.4**±1.8 | **90.1**±1.1 | **80.2**±1.7 | **77.8**±1.4 | **35.5**±2.2 |
| APPNP | 88.0±1.5 | 87.4±0.9 | 76.2±1.5 | 90.1±0.6 | 78.5±2.0 | 80.1±1.6 | 34.9±1.5 |
| GSL-APPNP | **89.0**±1.4 | **88.1**±2.1 | **77.0**±2.6 | **90.6**±0.9 | **79.9**±2.1 | **82.4**±2.5 | **36.0**±2.3 |
| PSL-APPNP | **89.0**±1.1 | **88.0**±2.6 | **77.0**±2.3 | 90.3±1.0 | 79.7±2.1 | 82.3±1.9 | **36.0**±1.6 |
| Graph-Heat | 86.6±1.4 | 87.1±1.6 | 74.8±2.5 | 89.2±1.1 | 75.8±2.0 | 79.3±1.6 | 34.0±1.6 |
| GSL-Heat | **87.4**±1.8 | **88.7**±2.5 | **76.6**±2.2 | **89.4**±1.3 | **77.4**±1.9 | **80.7**±2.1 | **35.6**±2.1 |
| PSL-Heat | 87.2±1.4 | **88.3**±1.8 | **76.7**±2.4 | 89.1±1.5 | 76.6±2.9 | 80.4±1.1 | 35.3±1.9 |
| BernNet | 88.2±0.7 | 88.4±1.3 | 76.6±0.7 | 94.0±1.2 | 86.3±1.9 | 85.8±1.4 | 36.0±1.6 |
| GSL-BernNet | **89.7**±1.6 | **89.5**±2.4 | **77.6**±2.0 | 94.1±1.0 | 88.3±2.4 | 86.8±1.7 | 37.3±2.2 |
| PSL-BernNet | **89.9**±1.3 | **89.7**±1.9 | 77.3±1.8 | **94.2**±1.0 | **88.4**±2.8 | **86.9**±1.2 | **37.5**±2.3 |
| GPRGNN | 87.1±0.9 | 87.9±1.1 | 76.7±1.5 | 92.8±1.2 | 85.2±1.7 | 86.2±2.1 | 36.3±1.5 |
| GSL-GPR | **88.1**±1.1 | 88.6±2.2 | 76.8±1.9 | **93.6**±1.0 | 86.1±1.8 | 87.8±2.0 | 36.7±2.4 |
| PSL-GPR | **88.4**±1.2 | **89.3**±1.6 | **77.4**±1.8 | 93.2±1.6 | **86.7**±1.5 | **88.2**±2.4 | **37.0**±2.5 |
| Jacobi | 88.7±1.4 | 88.0±0.7 | 76.7±1.8 | 93.6±1.3 | 87.4±1.8 | 84.2±2.1 | 36.5±1.7 |
| GSL-Jacobi | **89.0**±1.8 | **89.1**±2.2 | 77.7±2.2 | **94.4**±1.0 | **89.0**±2.3 | 86.3±2.0 | 37.7±2.0 |
| PSL-Jacobi | **89.0**±1.4 | **89.3**±1.9 | **77.8**±2.3 | 94.2±1.3 | 88.9±2.4 | **86.6**±1.1 | **37.8**±1.5 |

*Table 3.* Distinct eigenvalues statistics of normalized Laplacian on real-world datasets across multiple training runs using PSL-GNN models (95% confidence interval).

| Datasets | Cora | Pubmed | Citeseer | Photo | Texas | Cornell | Actor |
|---|---|---|---|---|---|---|---|
| $\|V\|$ | 2708 | 19717 | 3327 | 7650 | 183 | 183 | 7600 |
| $N_G$ | 2262 | 7647 | 1969 | 7511 | 73 | 60 | 4837 |
| $N_{PSL\text{-}GCN}$ | 2389±33 | 9031±45 | 2339±43 | 7604±19 | 147±11 | 142±16 | 6181±42 |
| $N_{PSL\text{-}APPNP}$ | 2408±31 | 8977±60 | 2300±41 | 7602±15 | 145±6 | 147±10 | 6234±37 |
| $N_{PSL\text{-}Heat}$ | 2376±25 | 8981±54 | 2301±37 | 7604±13 | 145±8 | 148±9 | 6225±40 |
| $N_{PSL\text{-}BernNet}$ | 2434±36 | 9018±56 | 2308±35 | 7609±10 | 144±13 | 149±6 | 6240±46 |
| $N_{PSL\text{-}GPR}$ | 2436±35 | 9024±57 | 2311±34 | 7603±19 | 145±12 | 148±7 | 6243±45 |
| $N_{PSL\text{-}Jacobi}$ | 2442±36 | 9030±58 | 2316±35 | 7567±30 | 147±13 | 150±8 | 6249±46 |

some works focus on establishing perturbation bounds of these two matrices. Dailey et al. (Dailey et al., 2014) conduct in-depth research on the perturbation bounds of symmetric positive semidefinite matrices and propose a strong relative perturbation bounds. For diagonally dominant matrices, a strong relative perturbation bound is proposed by Bai et al. (Ye, 2009). In this paper, we primarily focus on symmetric positive semidefinite and also diagonally dominant matrices, aiming to study the changes in their eigenvalues after perturbation.

## 8. CONCLUSION

In this paper, we propose a novel sheaf Laplacian by perturbing the normalized graph Laplacian, which enhances the spectral GNNs. Our approach addresses the limitations posed by repeated eigenvalues in the normalized graph Laplacian, thereby improving the expressive power of GNNs. Extensive experiments on benchmark datasets demonstrate the effectiveness of our method, showing notable improvements in node classification performance. Although our approach has been validated through both theoretical analysis and experiments, further exploration is required to fully reduce the multiplicity of eigenvalues and refine the matrix construction methods. Additionally, we have not yet investigated the impact of the perturbed sheaf Laplacian (PSL) on graph-level tasks, which remains an interesting direction for future research.

## Acknowledgements

We sincerely thank the anonymous reviewers for their constructive comments and thoughtful suggestions.

## Impact Statement

This paper presents work whose goal is to advance the field of Machine Learning. There are many potential societal consequences of our work, none which we feel must be specifically highlighted here.

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

# A. IMPORTANT NOTATIONS.

| Symbol | Definition |
| --- | --- |
| $G, \mathcal{F}$ | a graph, vector spaces over the graph |
| $V, E$ | set of nodes, set of edges |
| $A, D$ | adjacency matrix, degree matrix |
| $\mathcal{L}, \mathcal{L}_{\mathcal{F}}$ | normalized graph laplacian, sheaf laplacian |
| $\mathcal{L}_{block}, \overline{\mathcal{L}_{\mathcal{F}}}$ | block form of normalized graph laplacian, perturbed sheaf laplacian |
| $n, N, d$ | the dimension of $\mathcal{L}$ , the dimension of $\overline{\mathcal{L}_{\mathcal{F}}}$, the dimension of stalks |
| $\lambda, \tilde{\lambda}$ | the eigenvalue of $\mathcal{L}_{block}$, the eigenvalue of $\overline{\mathcal{L}_{\mathcal{F}}}$ |
| $X, \hat{X}$ | feature matrix (input matrix), the input matrix after the Fourier transform |
| $Y, Z$ | the output matrix after the Laplace transform, the final predication matrix |
| $P, Q$ | the applied perturbation matrix, the transport matrix (the transpose product of the restriction maps) |
| $g, \mathcal{R}$ | the filter function, the reshape operation |
| $\mathbf{u}$ | reflection vector |
| $\phi, \varphi$ | the minimum eigenvalue gap in $\mathcal{L}_{block}$, a function like multi-layer perceptrons |
| $\epsilon, \eta$ | the maximum perturbation ratio for every block, the upper limit of perturbations for all parameters in $\mathbf{u}$ |

# B. Proofs.

## B.1. Proof of Theorem 3.2

**Theorem B.1.** *When there are only $k$ distinct eigenvalues of the normalized Laplacian matrix, and the nonlinear real-valued function $\sigma$ satisfies $\sigma(x) \neq 0$ for $x \neq 0$, nonlinear spectral GNNs can produce at most $k$ different filter coefficients, and thus can only generate one-dimensional predictions with a maximum of $k$ arbitrary elements.*

***Proof.*** Let $Z = [z_1, \ldots, z_n]^T$ be the output, where $z_i \in \mathbb{R}$ and $z_i \neq 0$ for $i = 1, 2, \ldots, n$. Here, $k$ is the number of distinct eigenvalues of the normalized graph Laplacian matrix. Assume that the filter function is $g(\lambda)$, where $\lambda$ is the eigenvalue of the normalized graph Laplacian matrix. When $\lambda_i = \lambda_j$, $g(\lambda_i) = g(\lambda_j)$. Therefore, when the number of distinct eigenvalues of the Laplacian matrix is $k$, there are only $k$ different filter coefficients.

For the Fourier transform $Z = U \operatorname{diag}(g(\lambda))U^\top XW$, we assume that all elements of $X$ are non-negative. Let the vector $\mathbf{m} = U^\top XW$ and $\mathbf{q} = \operatorname{diag}(g(\lambda)) \cdot \mathbf{m}$. Then $Z = U\mathbf{q}$, where $Z \in \mathbb{R}^{n \times 1}$. If $\lambda_i = \lambda_j$, then $q_i = q_j \frac{m_i}{m_j}$. Since there are $k$ independent eigenvalues, there are only $k$ arbitrary elements in the vector $\mathbf{q}$. Therefore, any element in the vector $\mathbf{q}$ can be represented as a linear combination of these $k$ arbitrary elements, that is:

$$q_i = \alpha_1 q_{\ell_1} + \alpha_2 q_{\ell_2} + \cdots + \alpha_k q_{\ell_k}, \quad \forall i \in [1, n] \tag{13}$$

where $q_{\ell_1}, \ldots, q_{\ell_k}$ are the $k$ arbitrary elements in $\mathbf{q}$. Then we can use a matrix $U'$ with $n$ rows and $k$ columns, and a vector $\mathbf{q}'$ with $k$ elements to represent $Z$, i.e. $U'\mathbf{q}' = Z$. To illustrate this process, we group all eigenvectors according to their corresponding eigenvalues. For each distinct eigenvalue $\lambda_{\ell_s}$ (where $s = 1, 2, \ldots, k$), define the set: $S_s = \{j \mid \lambda_j = \lambda_{\ell_s}\}$.

$Z$ could be expressed as follow:

$$
\begin{bmatrix}
\sum_{j \in S_1} U_{1,j} \frac{m_j}{m_{\ell_1}} & \cdots & \sum_{j \in S_k} U_{1,j} \frac{m_j}{m_{\ell_k}} \\
\sum_{j \in S_1} U_{2,j} \frac{m_j}{m_{\ell_1}} & \cdots & \sum_{j \in S_k} U_{2,j} \frac{m_j}{m_{\ell_k}} \\
\vdots & \vdots & \vdots \\
\sum_{j \in S_1} U_{n,j} \frac{m_j}{m_{\ell_1}} & \cdots & \sum_{j \in S_k} U_{n,j} \frac{m_j}{m_{\ell_k}}
\end{bmatrix}
\begin{bmatrix}
q_{\ell_1} \\
q_{\ell_2} \\
\vdots \\
q_{\ell_k}
\end{bmatrix}
=
\begin{bmatrix}
z_1 \\
z_2 \\
\vdots \\
z_n
\end{bmatrix}
\tag{14}
$$

Since $U'$ has rank at most $k$, it spans a subspace of dimension at most $k$. Thus, $Z = U'\mathbf{q}'$ is confined within this subspace. We can choose up to $k$ linearly independent elements $z_{\ell_1}, z_{\ell_2}, \ldots, z_{\ell_k}$ as a basis. Therefore, any $z_i$ can be expressed as a linear combination:

$$
z_i = \beta_1 z_{\ell_1} + \beta_2 z_{\ell_2} + \cdots + \beta_k z_{\ell_k}, \quad \text{for } \beta_j \in \mathbb{R}
\tag{15}
$$

Let the activation function $\sigma(x)$ be a real-valued function, where $\sigma(x) \neq 0$ for $x \neq 0$. Since the elements in $Z$ are nonzero, let $Z' = \sigma(Z)$ and $\mathrm{rank}(U') = k$, then $z_i' = r_i \cdot z_i$, where $r_i \in \mathbb{R}$. According to Equation (15), we have:

$$
\begin{aligned}
z_i' &= \beta_1 \cdot \frac{r_i}{r_{\ell_1}} \cdot z_{\ell_1}' + \ldots + \beta_k \cdot \frac{r_i}{r_{\ell_k}} \cdot z_{\ell_k}' \\
&= \beta_1' \cdot z_{\ell_1}' + \ldots + \beta_k' \cdot z_{\ell_k}'
\end{aligned}
\tag{16}
$$

Therefore with the nonlinear activation function, spectral GNNs can still only generate at most $k$ arbitrary elements. □

### B.2. Proof of Theorem 4.1

**Theorem B.2.** *Let $P = \overline{\mathcal{L}_\mathcal{F}} - \mathcal{L}_{block}$ be the perturbation matrix applied to $\mathcal{L}_{block}$, and let $\phi = \min_{i,j;\, i \neq j} |\lambda_i - \lambda_j|$, where $\lambda_i$ are the eigenvalues of $\mathcal{L}_{block}$. If $\|P\|_2 < \frac{\phi}{2}$, then the eigenvalue multiplicity of $\mathcal{L}_\mathcal{F}$ will decrease.*

**Proof.** When the perturbation matrix $P$ is applied to $\mathcal{L}_{block}$, the new eigenvalues $\mu_i$ and $\mu_j$ of $\overline{\mathcal{L}_\mathcal{F}}$ corresponding to $\lambda_i$ and $\lambda_j$ will lie within a neighborhood of these eigenvalues. According to Weyl's inequality, we have: $|\mu_i - \lambda_i| \leq \|P\|_2$ and $|\mu_j - \lambda_j| \leq \|P\|_2$. This implies that $\mu_i$ lies in the interval: $[\lambda_i - \|P\|_2, \lambda_i + \|P\|_2]$, and $\mu_j$ lies in the interval: $[\lambda_j - \|P\|_2, \lambda_j + \|P\|_2]$.

Now, since $\phi = \min_{i,j;\, i \neq j} |\lambda_i - \lambda_j|$ and we assume $\|P\|_2 < \frac{\phi}{2}$, the length of each interval is $2\|P\|_2$, which is strictly less than $\phi$. The distance between the centers of these intervals is $\phi$, which is greater than the sum of the half-lengths of the intervals, $2\|P\|_2$. Thus, $[\lambda_i - \|P\|_2, \lambda_i + \|P\|_2]$ and $[\lambda_j - \|P\|_2, \lambda_j + \|P\|_2]$ do not overlap. This guarantees that the perturbed eigenvalues $\mu_i$ and $\mu_j$ remain distinct.

Since $\lambda_i$ and $\lambda_j$ represent the pair of eigenvalues with the smallest distance $\phi$, if their perturbation intervals do not overlap, then for any other pair of eigenvalues $\lambda_k$ and $\lambda_l$ (where $k \neq l$), their perturbation intervals will also not overlap due to $|\lambda_k - \lambda_l| \geq \phi$.

Therefore, no two perturbation intervals overlap, ensuring that the eigenvalue multiplicity of $\overline{\mathcal{L}_\mathcal{F}}$ will decrease as eigenvalues associated with higher multiplicities split into distinct values within their respective intervals. □

## C. Expressiveness of PSL-GNN

### C.1. Proof of Proposition 5.1

**Proposition C.1.** *The number of distinct eigenvalues of $\mathcal{L}_{block} \in \mathbb{R}^{nd \times nd}$ is the same as that of $\mathcal{L} \in \mathbb{R}^{n \times n}$, where $\mathcal{L}$ is the normalized graph Laplacian matrix and $\mathcal{L}_{block} = \mathcal{L} \otimes I$, with $I \in \mathbb{R}^{d \times d}$ being the identity matrix. Also, their corresponding GNNs, which share the same architecture, are mutually equivalent.*

**Proof.** We prove the first part of the proposition as follows. Let $\mathcal{L}$ have eigenvalues $\{\lambda_1, \lambda_2, \ldots, \lambda_n\}$, and let $I \in \mathbb{R}^{d \times d}$ be the identity matrix with eigenvalues $\{1, 1, \ldots, 1\}$. According to the eigenvalue property of the Kronecker product, the eigenvalues of $\mathcal{L}_{block} = \mathcal{L} \otimes I$ are $\lambda_i \cdot 1 = \lambda_i$ for each $i = 1, 2, \ldots, n$. Since $I$ only introduces eigenvalues of 1, the

Kronecker product does not add new eigenvalues. Each eigenvalue $\lambda_i$ of $\mathcal{L}$ is repeated $d$ times, meaning the number of distinct eigenvalues of $\mathcal{L}_{\text{block}}$ is the same as that of $\mathcal{L}$.

Consider an input matrix $\mathbf{X} \in \mathbb{R}^{n \times f}$, where $f = d \times k$. Reshape $\mathbf{X}$ into $\mathbf{X}' \in \mathbb{R}^{nd \times k}$. Let $\mathbf{Y} = \mathcal{L}\mathbf{X}$ and $\mathbf{Y}' = \mathcal{L}_{\text{block}}\mathbf{X}'$. We partition $\mathbf{X}$ into $n$ row blocks: $\mathbf{X} = \left[(\mathbf{X}^{(1)})^\top, \ldots, (\mathbf{X}^{(n)})^\top\right]^\top$, where each $\mathbf{X}^{(i)} \in \mathbb{R}^{1 \times f}$. Reshape each $\mathbf{X}^{(i)}$ into a block matrix $\mathbf{X}'^{(i)} \in \mathbb{R}^{d \times k}$. The reshaped matrix $\mathbf{X}' \in \mathbb{R}^{nd \times k}$ is formed by stacking these blocks.

Now compute $\mathbf{Y}' = \mathcal{L}_{\text{block}}\mathbf{X}'$. The Kronecker product $\mathcal{L}_{\text{block}} = \mathcal{L} \otimes I_d$ implies that each element $\mathcal{L}_{ij}$ of $\mathcal{L}$ is multiplied by the $d \times d$ identity matrix $I_d$. Thus, $\mathbf{Y}'$ can be expressed as: $\mathbf{Y}' = \left[\sum_{j=1}^{n} \mathcal{L}_{1j}\mathbf{X}'^{(j)} \quad \cdots \quad \sum_{j=1}^{n} \mathcal{L}_{nj}\mathbf{X}'^{(j)}\right]^\top$.

Next, we reshape $\mathbf{Y}'$ back to $\mathbf{Y}'_{\text{reshaped}} \in \mathbb{R}^{n \times f}$. For each $i, s, t$, the element $\mathbf{Y}_{i,(s-1)d+t}$ in $\mathbf{Y}$ corresponds to $\mathbf{Y}'^{(i)}_{s,t}$ in $\mathbf{Y}'$. Thus, we have: $\mathbf{Y}_{i,(s-1)d+t} = \mathbf{Y}'_{\text{reshaped}}(i, (s-1)d+t)$.

Since this holds for all $i, s, t$, we conclude that $\mathbf{Y} = \mathbf{Y}'_{\text{reshaped}}$. $\qquad\square$

### C.2. Proof of Corollary 5.2

**Corollary C.2.** *PSL-GNN has stronger one-dimensional prediction ability.*

**Proof.** Consider a block form matrix $\mathcal{L}_{block}$ and its corresponding perturbed sheaf Laplacian matrix $\overline{\mathcal{L}_\mathcal{F}}$. Let the input matrix be $X \in \mathbb{R}^{n \times 1}$. Their outputs are given as $Z_1 = U_1 g(\lambda) U_1^\top X W_1$ and $Z_2 = U_2 g(\lambda) U_2^\top X W_2$, respectively. Define vectors $\mathbf{m}_1 = X W_1$ and $\mathbf{m}_2 = X W_2$, and let $\mathbf{q}_1 = g(\lambda) \cdot \mathbf{m}_1$ and $\mathbf{q}_2 = g(\lambda) \cdot \mathbf{m}_2$.

Suppose the number of distinct eigenvalues of $\overline{\mathcal{L}_\mathcal{F}}$ and $\mathcal{L}$ are $k_1$ and $k_2$, respectively. According to the Proposition 5.1, the number of distinct eigenvalues of $\mathcal{L}_{\text{block}}$ is equal to that of $\mathcal{L}$, so the number of distinct eigenvalues of $\mathcal{L}_{\text{block}}$ is $k_2$. When $\lambda_i = \lambda_j$, $g(\lambda_i) = g(\lambda_j)$.

Therefore, the outputs can be expressed as $Z_1 = U_1' \mathbf{q}_1'$ and $Z_2 = U_2' \mathbf{q}_2'$, where $U_1'$ is an $n \times k_1$ matrix, $U_2'$ is an $n \times k_2$ matrix, and $\mathbf{q}_1'$ and $\mathbf{q}_2'$ are vectors with $k_1$ and $k_2$ elements, respectively.

Since the ranks of $U_1'$ and $U_2'$ are at most $k_1$ and $k_2$, $Z_1$ and $Z_2$ each have at most $k_1$ and $k_2$ arbitrary elements, respectively. Given that $k_1 > k_2$, the number of arbitrary elements in $Z_1$ is greater than that in $Z_2$. $\qquad\square$

### C.3. Proof of Corollary 5.3

**Corollary C.3.** *PSL-GNN loses fewer frequency components.*

**Proof.** Consider a block form matrix $\mathcal{L}_{block}$ and its corresponding perturbed sheaf Laplacian matrix $\overline{\mathcal{L}_\mathcal{F}}$. Let the maximum multiplicity of the eigenvalues of $\mathcal{L}_{block}$ be $m$, and the maximum multiplicity of the eigenvalues of $\overline{\mathcal{L}_\mathcal{F}}$ be $\hat{m}$. Since $\hat{m} < m$, PSL-GNN is able to lose fewer frequency components even in the worst-case design of the filtering function. $\quad\square$

### C.4. Proof of Corollary 5.4

**Corollary C.4.** *PSL-GNN has a stronger capability in processing frequency components.*

**Proof.** Consider a block form matrix $\mathcal{L}_{block}$ and its corresponding perturbed sheaf Laplacian matrix $\overline{\mathcal{L}_\mathcal{F}}$. Let the number of distinct eigenvalues of $\mathcal{L}_{block}$ and $\overline{\mathcal{L}_\mathcal{F}}$ be $k_1$ and $k_2$, respectively. Since $k_1 < k_2$, the PSL-GNN filter has more coefficients, implying that it can handle frequency components in a more diverse manner. $\qquad\square$

*Remark* C.5. We explain Corollary 5.4 more clearly: the eigenvalues of $\mathcal{L}_{\text{block}}$ are divided into $k_1$ groups, while those of $\mathcal{L}_\mathcal{F}$ are divided into $k_2$ groups, with $k_2 > k_1$. Accordingly, the filter coefficients are divided into $k_1$ and $k_2$ groups, with identical coefficients within each group. This means that the frequency components associated with each eigenvalue in a group are scaled by the same factor. Since $k_2 > k_1$, PSL-GNN can independently scale more frequency components.

## D. Perturbation Bounds of Eigenvalues

**Theorem D.1.** *Let $\mathcal{L}_{block}$ and $\mathcal{L}_\mathcal{F} = \mathcal{L}_{block} + P$ be $N \times N$ symmetric positive semidefinite matrices, with eigenvalues $\lambda_1, \ldots, \lambda_N$ and $\tilde{\lambda}_1, \ldots, \tilde{\lambda}_N$, respectively, where $\mathcal{L}_{block}$ is diagonally dominant. Suppose the perturbation matrix $P$ satisfies $|P_{ij}| \le \epsilon |(L_{block})_{ij}|$ for all $i \ne j$ for all $i$, where $0 < \epsilon < \frac{1}{2}$. Hence, for each $i$, $|\tilde{\lambda}_i - \lambda_i| \le \frac{2\epsilon}{1-2\epsilon} \tilde{\lambda}_{\max}$.*

**Proof.** First, we bound the spectral norm of $P$. For each row $i$, we have:

$$\sum_{j=1}^{n} |P_{ij}| = |P_{ii}| + \sum_{j \neq i} |P_{ij}| \leq \epsilon v_i + \epsilon \sum_{j \neq i} |(\mathcal{L}_{\text{block}})_{ij}| = \epsilon \left( v_i + \sum_{j \neq i} |(\mathcal{L}_{\text{block}})_{ij}| \right) = \epsilon (\mathcal{L}_{\text{block}})_{ii}. \tag{17}$$

Therefore:

$$\|P\|_{\infty} \leq \epsilon \max_{1 \leq i \leq n} (\mathcal{L}_{\text{block}})_{ii}. \tag{18}$$

Since $\mathcal{L}_{\text{block}}$ is diagonally dominant and positive semidefinite:

$$\|\mathcal{L}_{\text{block}}\|_{\infty} = \max_{1 \leq i \leq n} \sum_{j=1}^{n} |(\mathcal{L}_{\text{block}})_{ij}| \leq 2 \max_{1 \leq i \leq n} (\mathcal{L}_{\text{block}})_{ii} \leq 2\|\mathcal{L}_{\text{block}}\|_2, \tag{19}$$

where $\|\mathcal{L}_{\text{block}}\|_2$ denotes the spectral norm of $\mathcal{L}_{\text{block}}$. Thus:

$$\|P\|_2 \leq \|P\|_{\infty} \leq \epsilon \|\mathcal{L}_{\text{block}}\|_{\infty} \leq 2\epsilon \|\mathcal{L}_{\text{block}}\|_2. \tag{20}$$

Next, we apply Weyl's theorem, which states that for symmetric matrices $\mathcal{L}_{\text{block}}$ and $P$: $|\tilde{\lambda}_i - \lambda_i| \leq \|P\|_2$. Substituting the norm bound derived above:

$$|\tilde{\lambda}_i - \lambda_i| \leq \|P\|_2 \leq 2\epsilon \|\mathcal{L}_{\text{block}}\|_2. \tag{21}$$

We now relate $\|\mathcal{L}_{\text{block}}\|_2$ to $\tilde{\lambda}_i$. Since $\mathcal{L}_{\mathcal{F}} = \mathcal{L}_{\text{block}} + P$:

$$\|\mathcal{L}_{\text{block}}\|_2 \leq \|\mathcal{L}_{\mathcal{F}}\|_2 + \|P\|_2 \leq \|\mathcal{L}_{\mathcal{F}}\|_2 + 2\epsilon \|\mathcal{L}_{\text{block}}\|_2. \tag{22}$$

Rearranging gives:

$$\|\mathcal{L}_{\text{block}}\|_2 \leq \frac{\|\mathcal{L}_{\mathcal{F}}\|_2}{1 - 2\epsilon}. \tag{23}$$

Given that $\|\mathcal{L}_{\mathcal{F}}\|_2 = \tilde{\lambda}_{\max} \geq \tilde{\lambda}_i$, it follows that:

$$|\tilde{\lambda}_i - \lambda_i| \leq 2\epsilon \|\mathcal{L}_{\text{block}}\|_2 \leq \frac{2\epsilon}{1 - 2\epsilon} \|\mathcal{L}_{\mathcal{F}}\|_2 = \frac{2\epsilon}{1 - 2\epsilon} \tilde{\lambda}_{\max}. \tag{24}$$

Thus, the bound is proven. $\square$

# E. Experimental Setup

## E.1. Experimental Details

We construct every reflection matrix $\mathcal{F}_{i \trianglelefteq (i,j)}$ based on the method in (Obukhov, 2021). For downstream task, we update the perturbed sheaf Laplacian matrix every 10 epochs. Given its high dimensionality, we employ sparse matrix storage to reduce computational overhead. Additionally, we apply an early stopping mechanism with a maximum of 1500 epochs and a patience threshold of 100. For each PSL-GNN and its corresponding GNN, we train them for the same number of times and take the average of their respective accuracies as the final result for each model. For model training, we use the Adam optimizer to optimize all models on an NVIDIA GeForce RTX 4090 GPU.

## E.2. Datasets

We use seven benchmark datasets, categorized as follows: (1) Citation Networks: Cora (McCallum et al., 2000), Citeseer (Giles et al., 1998), and Pubmed (Sen et al., 2008) are citation networks with nodes as publications, edges as citation links, and labels indicating topics or fields; (2) Co-purchase Networks: Photo (Shchur et al., 2018) is a co-purchase network with nodes as products, edges as co-purchases, and labels categorizing products; (3) Webpage Networks: Texas and Cornell

(Shchur et al., 2018) are webpage networks with nodes as webpages, edges as hyperlinks, and labels indicating categories; (4) Actor Co-occurrence Network: Actor (Shchur et al., 2018) is a co-occurrence network with nodes as actors, edges as co-occurrences, and labels classifying actors.

The information of each dataset is shown in Table 5. Additionally, due to the diagonal dominance of the normalized graph Laplacian $\mathcal{L}$, its eigenvalue $\lambda = 1$ has higher multiplicity. Thus we also compute the proportion $P_{\lambda=1}$ of this multiplicity for analysis.

Table 5. The statistical information of the datasets.

| Datasets | Cora | Pubmed | Citeseer | Photo | Texas | Cornell | Actor |
|---|---|---|---|---|---|---|---|
| Nodes | 2708 | 19717 | 3327 | 7650 | 183 | 183 | 7600 |
| Edges | 5278 | 44324 | 4552 | 245861 | 287 | 278 | 26705 |
| Features | 1433 | 500 | 3703 | 745 | 1703 | 1703 | 932 |
| Classes | 7 | 3 | 6 | 8 | 5 | 5 | 5 |
| $P_{\lambda=1}$ | 0.125 | 0.639 | 0.171 | 0.048 | 0.362 | 0.349 | 0.155 |

### E.3. GNN Baselines

To evaluate the performance of our proposed method, we choose six different spectral GNNs as follows.

GCN (Kipf & Welling, 2016) is a spectral graph neural network that uses a mean aggregator to combine information from neighboring nodes, bridging the gap between spectral and spatial GNN methods. APPNP (Klicpera et al., 2018) employs personalized PageRank to propagate node features, addressing the oversmoothing problem in deep networks. Graph-Heat (Xu et al., 2019a) incorporates the heat kernel for graph signal filtering, providing a natural mechanism to smooth or diffuse signals over the graph. BernNet (He et al., 2021) improves spectral graph convolutions using Bernstein polynomials, offering better flexibility to handle varying graph structures. GPRGNN (Chien et al., 2021) generalizes PageRank algorithms to enhance performance on graphs with heterophilic structures. Lastly, JacobiConv (Wang & Zhang, 2022) leverages orthogonal Jacobi polynomials to enhance expressiveness and optimization in spectral graph neural networks.

For all baseline models, we use either their official implementations or implementations provided by PyTorch Geometric to ensure fair comparisons.

### E.4. Hyper-Parameters

To ensure a fair comparison, we keep all hyper-parameters the same for each PSL-GNN and its corresponding baseline GNN, except for $\eta$ and $d$. Specifically, for all models, we set the learning rate to 0.05, weight decay to $5e-4$, and the number of hidden units to 64. We set the order of the polynomial filters—BernNet, APPNP, Graph-Heat, and Jacobi—to 10. For all PSL-GNN models, we search $\eta$ within $\{1e-1, 1e-2, 1e-3, 1e-4\}$ and $d$ within $\{2, 3, 4\}$ to achieve the best model performance.

## F. Additional Experimental Studies

### F.1. Complexity Analysis

We first analyze the space and time costs of constructing the perturbed sheaf Laplacian, then the per-epoch cost of PSL-GCN (with and without PSL reconstruction), and finally report empirical runtimes.

#### F.1.1. SPACE COMPLEXITY & PARAMETERS.

Each restriction map uses $d$ free parameters, so forming the edge-incidence matrix $\mathcal{B}_{\mathcal{F}}$ (with two entries per edge) requires space $= O\big(2E \cdot d^2 + n \cdot d\big)$,

#### F.1.2. PSL CONSTRUCTION TIME.

We build the signed incidence matrix $\mathcal{B}_{\mathcal{F}}$ (size $nd \times \mathrm{nnz}_{\mathcal{B}}$), then compute $\overline{\mathcal{L}_{\mathcal{F}}} = \mathcal{B}_{\mathcal{F}} \mathcal{B}_{\mathcal{F}}^T$. Since $\mathcal{B}_{\mathcal{F}}$ has $\mathrm{nnz}_{\mathcal{B}}$ nonzeros, this sparse-sparse product costs $O\big(\mathrm{nnz}_{\mathcal{B}}^2\big)$.

F.1.3. PER-EPOCH COST OF PSL-GCN.

Every 10 epochs we rebuild the PSL; thus per-epoch we either include or omit the $O(\mathrm{nnz}_{\mathcal{B}}^2)$ term:

- **Including PSL construction:** $O\big(\mathrm{nnz}_{\mathcal{B}}^2\big) + O\big(n\,f_1\,d\,f_3\big) + O\big(2\,n\,d\,f_3\,(r+f_4)\big) + O\big(n\,d\,f_{\mathrm{last}}\,c\big).$

- **Amortized (excluding PSL construction):** $O\big(n\,f_1\,d\,f_3\big) + O\big(2\,n\,d\,f_3\,(r+f_4)\big) + O\big(n\,d\,f_{\mathrm{last}}\,c\big).$

F.1.4. EMPIRICAL RUNTIMES.

Table 6 reports average epoch time (ms) / total runtime per fold (s) for PSL-GCN vs. Neural Sheaf Diffusion (Diag-NSD, O(d)-NSD, Gen-NSD) (Bodnar et al., 2022):

*Table 6.* Average epoch time (ms) / total runtime per fold (s).

| Dataset | Cora | Pubmed | Citeseer | Photo | Texas | Cornell | Actor |
|---|---|---|---|---|---|---|---|
| Diag-NSD | 26.4/5.9 | 124.3/34.7 | 25.2/5.8 | 90.6/17.4 | 8.3/1.6 | 8.1/1.5 | 98.4/21.5 |
| O(d)-NSD | 57.0/12.3 | 204.5/66.3 | 64.1/13.6 | 143.5/26.3 | 26.8/6.7 | 27.4/7.2 | 128.2/31.9 |
| Gen-NSD | 85.3/17.6 | 231.1/74.3 | 92.4/20.6 | 166.7/34.6 | 34.3/14.4 | 31.4/14.2 | 177.6/45.2 |
| PSL-GCN | 16.0/4.6 | 83.7/17.4 | 18.3/5.2 | 65.3/15.8 | 6.3/1.1 | 6.4/1.1 | 72.7/17.3 |

### F.2. Comparison of Spectral Gap

The Laplacian spectrum measures graph connectivity. Cheeger's inequality $2h_G > \lambda_1 > \frac{h_G^2}{2}$ shows that a larger spectral gap $\lambda_1$ implies better connectivity (Jamadandi et al., 2024). The Cheeger constant $h_G$ quantifies the tightest connectivity bottleneck in a graph. In (Lu et al., 2024), the new operator's (**H**) eigenvalues are defined as $u_i = \beta\lambda_i + (1-\beta)v_i$, where $v_i = 2i/n - 2$. When $v_1 < \lambda_1$, it follows that $u_1 < \lambda_1$ (because $u_1 - \lambda_1 = (1-\beta)(v_1 - \lambda_1)$ with $\beta < 1$), which reduces the spectral gap and impairs information flow, especially in large graphs, where $v_1 = 2/n - 2$ is very, very small. However, our approach perturbs the normalized Laplacian in a way that maintains subtle eigenvalue differences regardless of graph size, thereby preserving the topological information.

Below, we verify our argument above by presenting the spectral gaps' orders of magnitude for the normalized graph Laplacian and PSL across various datasets. As shown in the Table 7, for each dataset, $v_1 < \lambda_1$, $u_1 < \lambda_1$ , and $\lambda_1 \leq \hat{\lambda}_1$, which indicates that Lu et al.'s approach (Lu et al., 2024) indeed reduces the spectral gap, thus compromising the original topological information encoded in the normalized Laplacian matrix.

*Table 7.* Comparison of spectral gap

| | Cora | Pubmed | Citeseer | Photo | Texas | Cornell | Actor |
|---|---|---|---|---|---|---|---|
| $n$ | 2708 | 19717 | 3327 | 7650 | 183 | 183 | 7600 |
| $v_1$ | 7e-4 | 1e-4 | 6e-4 | 2e-4 | 1e-2 | 1e-2 | 2e-4 |
| $\lambda_1$ | 4e-3 | 1e-2 | 1e-3 | 1e-3 | 5e-2 | 7e-2 | 3e-2 |
| $\hat{\lambda}_1$ | 5e-3 | 1e-2 | 2e-3 | 1e-3 | 5e-2 | 7e-2 | 3e-2 |

### F.3. Node Classification in A Fully-Supervised Split

We train all models in a fully-supervised split (60% / 20% / 20%). For each baseline (including the EC-GNN variants), we adopt the best hyperparameter settings reported in the original paper. For all PSL-GNN variants, we use the same settings as their corresponding baselines, with $\eta = 1 \times 10^{-3}$ and $d = 2$. The results are summarized in Table 8 and Table 9.

*Table 8.* Results on real-world datasets in the dense splitting (60% / 20% / 20%).

| Datasets | Cora | Citeseer | PubMed | Computers | Photo |
|---|---|---|---|---|---|
| GCN | 87.16±1.03 | 79.84±0.73 | 86.79±0.32 | 83.31±1.01 | 88.24±0.78 |
| EC-GCN | 88.73±0.89 | 80.63±1.01 | 87.68±0.83 | 83.94±1.17 | 90.45±1.20 |
| PSL-GCN | **89.09±1.14** | **81.04±0.63** | **88.75±1.24** | **84.12±1.65** | **91.48±0.81** |
| APPNP | 88.23±1.05 | 80.61±0.94 | 88.17±0.38 | 85.35±0.38 | 90.48±0.57 |
| EC-APPNP | 89.54±1.43 | 81.48±1.23 | 88.42±1.36 | 86.10±1.07 | 90.75±1.06 |
| PSL-APPNP | **90.91±1.14** | **81.92±1.36** | **88.88±1.42** | **86.43±0.68** | **91.82±0.96** |
| Graph-Heat | 87.34±1.41 | 78.82±1.26 | 87.52±0.89 | 83.23±1.37 | 89.97±0.82 |
| EC-Heat | 88.13±1.02 | 79.44±1.12 | 88.93±0.91 | 83.86±1.24 | 90.57±0.86 |
| PSL-Heat | **88.51±1.38** | **80.02±1.31** | **89.21±1.57** | **84.64±1.58** | **90.87±0.91** |
| BernNet | 88.48±1.14 | 80.08±1.07 | 88.73±1.42 | 87.68±0.53 | 94.31±0.86 |
| EC-BernNet | 88.64±0.55 | 80.30±0.98 | 89.07±1.37 | 88.34±1.03 | 94.50±1.01 |
| PSL-BernNet | **90.01±0.93** | **80.86±1.15** | **90.13±1.44** | **88.74±0.56** | **94.79±1.03** |
| GPRGNN | 88.54±0.82 | 80.09±1.03 | 88.52±0.46 | 87.01±0.74 | 93.87±0.34 |
| EC-GPR | 89.41±0.69 | 80.66±1.01 | 89.64±0.53 | 89.91±0.68 | 94.76±1.02 |
| PSL-GPR | **90.13±0.92** | **81.11±0.76** | **89.82±1.89** | **89.93±1.70** | **94.87±0.96** |
| Jacobi | 88.96±0.68 | 80.73±0.88 | 89.67±0.82 | 90.42±0.31 | 95.52±0.33 |
| EC-Jacobi | 89.06±0.67 | 81.28±0.96 | 89.87±0.42 | 90.33±0.28 | 95.54±0.36 |
| PSL-Jacobi | **90.73±1.34** | **81.54±1.12** | **90.42±1.13** | **90.83±0.61** | **95.69±0.52** |

*Table 9.* Results on real-world datasets in the dense splitting (60% / 20% / 20%).

| Datasets | Chameleon | Actor | Squirrel | Texas | Cornell |
|---|---|---|---|---|---|
| GCN | 59.65±2.52 | 34.63±1.62 | 46.64±1.28 | 78.65±2.98 | 76.42±5.21 |
| EC-GCN | 61.81±1.94 | 36.12±1.44 | 50.31±0.84 | 80.72±2.39 | 77.38±4.37 |
| PSL-GCN | **62.04±2.71** | **36.74±1.43** | **52.73±1.54** | **81.45±2.43** | **78.13±4.32** |
| APPNP | 51.78±1.91 | 39.59±0.83 | 34.74±0.60 | 90.89±1.53 | 91.84±2.12 |
| EC-APPNP | 52.93±2.12 | 40.32±0.94 | 35.33±0.84 | 91.32±1.34 | 91.97±1.64 |
| PSL-APPNP | **54.49±1.74** | **40.67±0.91** | **35.63±0.58** | **91.53±1.49** | **92.07±1.94** |
| Graph-Heat | 63.46±0.92 | 35.74±1.31 | 44.35±2.14 | 80.06±0.94 | 76.87±0.85 |
| EC-Heat | 64.86±1.22 | 36.51±0.94 | 45.01±1.87 | 81.25±1.36 | 77.91±0.63 |
| PSL-Heat | **65.60±1.23** | **36.93±1.45** | **45.32±1.89** | **81.49±1.10** | **80.94±1.05** |
| BernNet | 68.25±1.59 | 41.82±0.99 | 51.37±0.75 | 94.14±1.67 | 92.16±1.77 |
| EC-BernNet | 74.20±1.33 | 41.87±1.02 | 62.79±0.78 | 94.37±1.44 | 93.77±1.26 |
| PSL-BernNet | **75.09±1.46** | **41.93±1.30** | **63.82±1.33** | **94.54±1.80** | **94.01±0.93** |
| GPRGNN | 67.14±1.10 | 39.92±0.65 | 50.08±1.95 | 92.97±1.41 | 91.32±2.02 |
| EC-GPR | 74.24±1.06 | 40.42±0.77 | 62.48±2.03 | 92.27±1.92 | 90.79±2.22 |
| PSL-GPR | **74.78±1.34** | **41.17±0.96** | **63.65±0.87** | **94.74±1.65** | **92.44±1.85** |
| Jacobi | 74.23±1.45 | 41.16±0.70 | 57.38±1.24 | 93.45±2.03 | 92.94±2.38 |
| EC-Jacobi | 75.64±1.51 | 41.01±0.74 | 59.87±0.91 | 93.48±1.49 | 93.29±2.33 |
| PSL-Jacobi | **75.87±1.44** | **41.94±0.75** | **61.47±0.98** | **94.20±1.75** | **93.91±1.96** |

From the above results, PSL-GNNs consistently outperform the baselines, which validates the effectiveness of the perturbed sheaf Laplacian (PSL).

### F.4. Eigenvalue Perturbation Study

Every restriction map $\mathcal{F}_{v \trianglelefteq e}$ is generated using $d$ parameters, and is bounded by $\eta$, where $d$ represents the dimension of the stalks. To evaluate the impact of different perturbation scales $\eta$ on the eigenvalues, we compare the maximum relative eigenvalue perturbation between the resulting sheaf Laplacian matrix and the normalized graph Laplacian matrix. The

maximum relative perturbation is defined as:

$$\Delta\lambda = \max \left| \frac{\hat{\lambda}_i - \lambda_i}{\lambda_i} \right|, \quad \text{for } i = 1, 2, \dots, N,$$

(25)

where N is the dimension of the matrix, $\hat{\lambda}_i$ denotes the eigenvalues of the sheaf Laplacian matrix, and $\lambda_i$ represents the

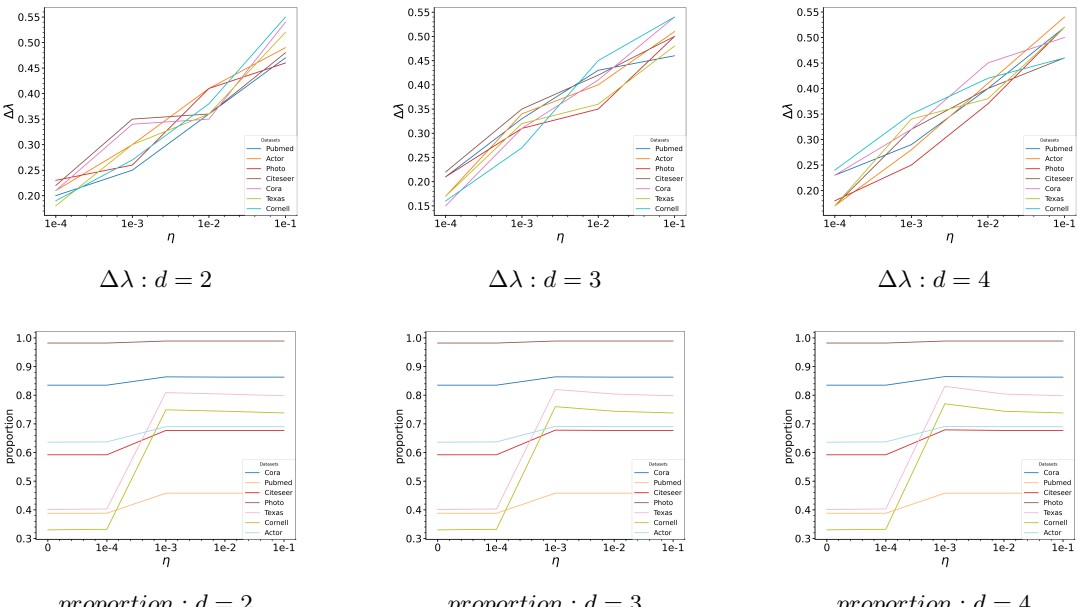

*Figure 4.* The relative eigenvalue perturbations and the proportion of distinct eigenvalues of the perturbed sheaf Laplacian across different datasets under varying $\eta$ and $d$.

eigenvalues of the normalized graph Laplacian matrix. We also examine how the proportion of distinct eigenvalues, defined as k/|v| (where k is the number of distinct eigenvalues), varies with different $\eta$. The results for each dataset are shown in the Figure 4.

As shown in Figure 4, when $\eta$ decreases, $\Delta\lambda$ also decreases, which is consistent with the theoretical analysis for the perturbation of eigenvalues in Theorem 5.5. At $\eta = 1e-3$, the number of distinct eigenvalues peaks but decreases for $\eta > 1e-3$, as larger perturbations may cause eigenvalues to overlap, violating Theorem 4.1. Furthermore, the impact of $d$ is negligible.

## F.5. Visualization of Eigenvalues Distribution

To better illustrate whether the multiplicities of eigenvalues in the perturbed sheaf Laplacian $\overline{\mathcal{L_F}}$ are reduced across different datasets, we visualize its eigenvalue distribution alongside that of the normalized graph Laplacian $\mathcal{L}$. Figure 5 shows the eigenvalues distributions under a perturbation scale of $\eta = 1e-3$, where the vertical axis (density) represents the proportion of each distinct eigenvalue relative to the total, and the horizontal axis ($\lambda$) represents the eigenvalues. Based on the analysis of these figures, we observe the following:

1. For almost every graph dataset, the perturbed sheaf Laplacian matrix consistently exhibits fewer repeated eigenvalues compared to the corresponding normalized graph Laplacian matrix.

2. On small graph datasets, the reduction in eigenvalue multiplicity is more pronounced, which aligns with the greater average performance gains observed in Table 1.

3. For every graph dataset, the eigenvalue 1 always has the highest multiplicity. This explains why GCN performs worse than other baselines, as its filtering function removes frequency components associated with eigenvalue 1. In contrast, $\mathcal{L_F}$ reduces the multiplicity of eigenvalue 1, which explains the highest average improvement for GCN, as it preserves the frequency components.

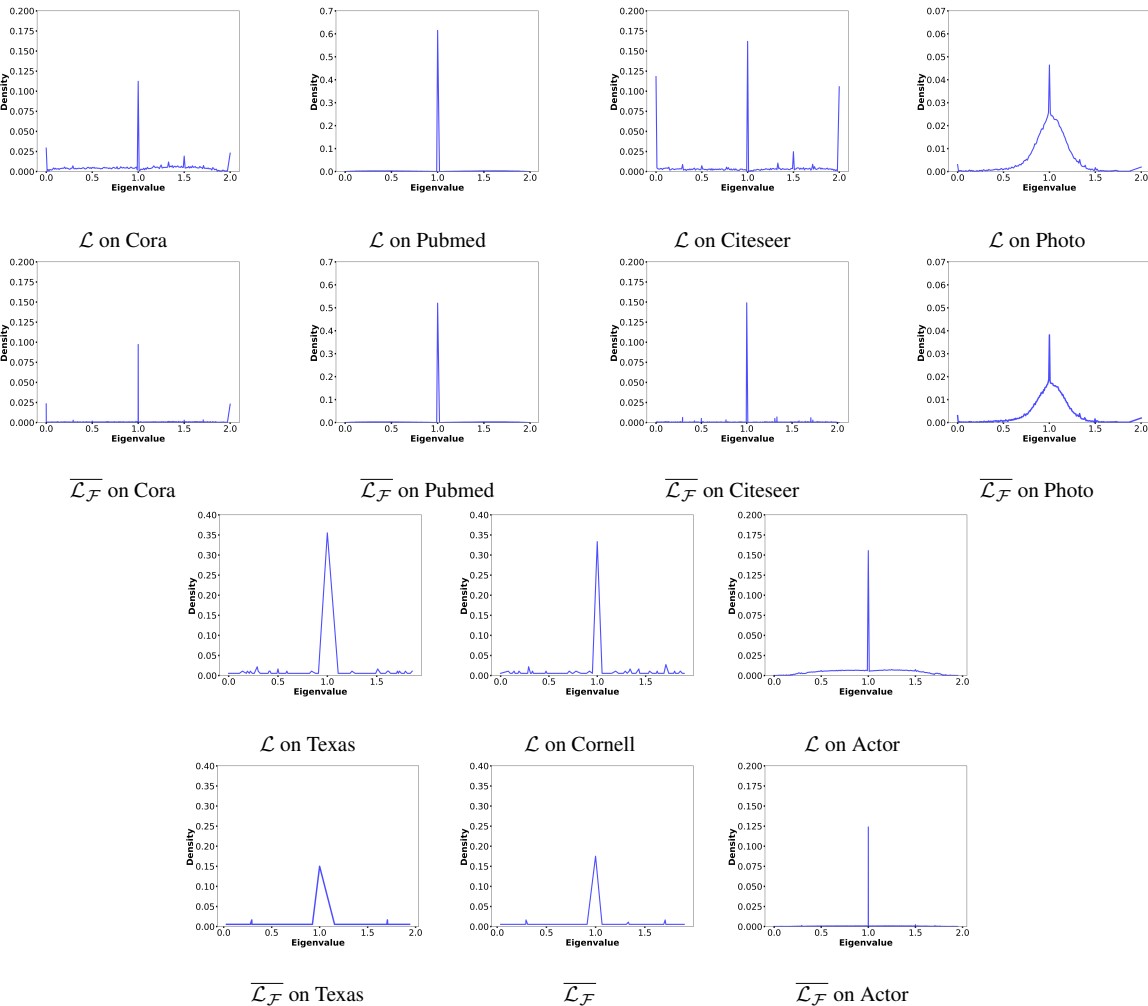

*Figure 5.* Eigenvalues distribution of normalized graph Laplacian matrices and perturbed sheaf Laplacian matrices.

## F.6. Hyper-Parameters Study

To demonstrate how hyper-parameters influence the performance of the model, we study the effect of the two main hyper-parameters of PSL-GPF and PSL-GCN, i.e. perturbation size $\eta$ and stalk dimension $d$. For the remaining parameters, we set the same configuration across Cora, Texas, and Actor.

The results are shown in the Figure 6. From the results, we have the following observations:

1. These models perform best when $\eta = 1e - 3$ and $\eta = 1e - 2$. This could be explained by Figure 5: when $\eta = 1e - 3$, the proportion of distinct eigenvalues in the fixed perturbed sheaf Laplacian matrix is maximized. Parameters of the restriction mapping are constrained within the range $[1e - 4, 1e - 3]$, and the parameters might gradually approach the upper bound of $1e - 3$. When $\eta = 1e - 2$, the performance is similar, as the parameters are constrained within the range $[1e - 3, 1e - 2]$, they might gradually approach the lower bound of $1e - 3$.

2. When $\eta = 1e - 4$, the model's performance is similar to that when no perturbation is applied (i.e., $\eta = 0$). As shown in Figure 5, the perturbation effect is nearly negligible in this case, and the perturbed sheaf Laplacian matrix becomes almost identical to the normalized graph Laplacian matrix.

## F.7. Comparison between PSL and GSL

In the fully-supervised split, we compared the performance of PSL and GSL. The results are shown below:

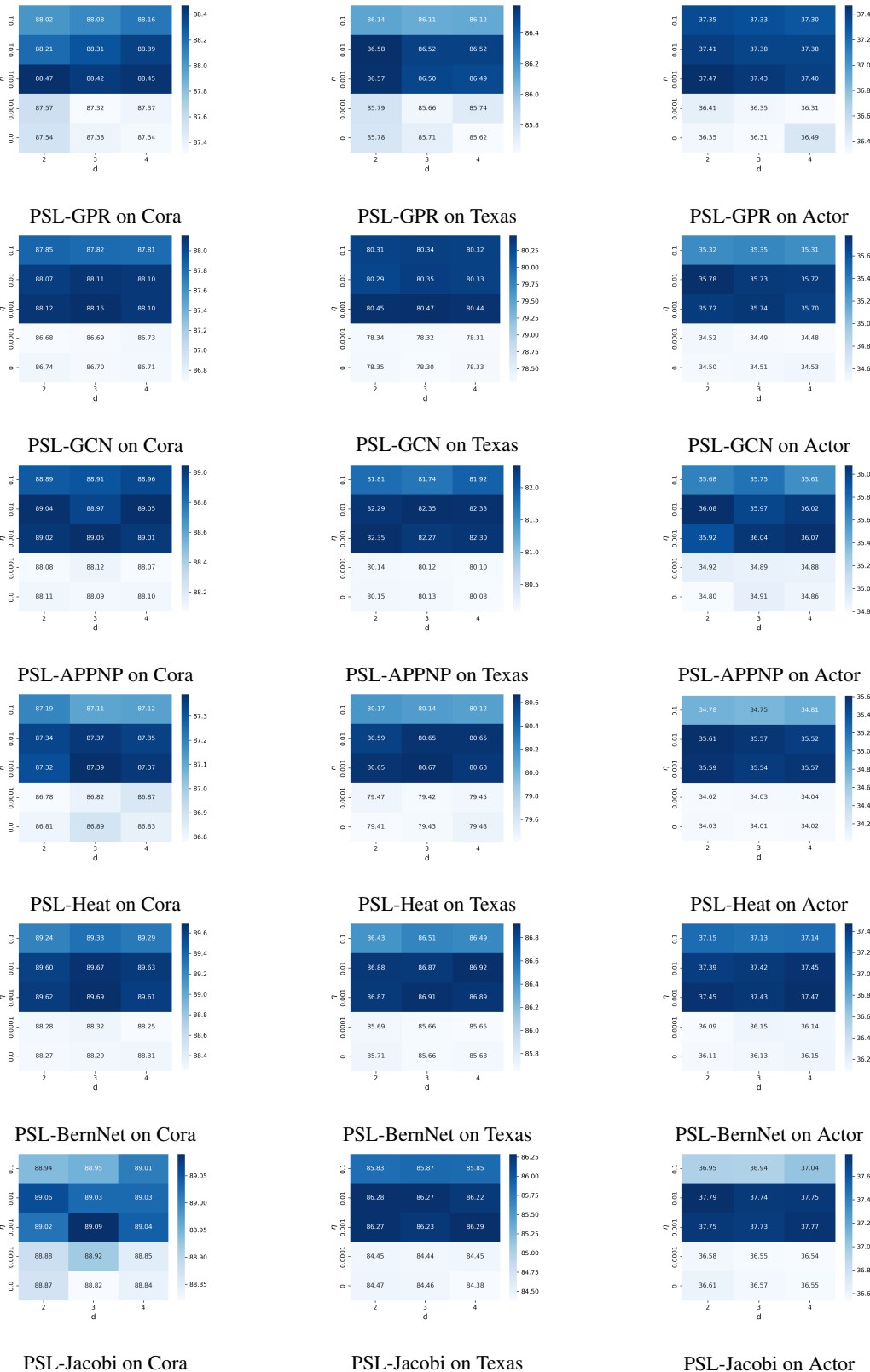

*Figure 6.* Hyper-Parameters Study.

*Table 10.* Comparison of models based on three Laplacian matrices (GNN vs. GSL-GNN vs. PSL-GNN): Mean accuracy (%) ±95 (%) on seven datasets (in dense splitting: 60% / 20% / 20%).

| Datasets | Cora | Pubmed | Citeseer | Photo | Texas | Cornell | Actor |
|---|---|---|---|---|---|---|---|
| GCN | 87.16±1.03 | 86.79±0.32 | 79.84±0.73 | 88.24±0.78 | 78.65±2.98 | 76.42±5.21 | 34.63±1.62 |
| GSL-GCN | 88.73 ±0.98 | 87.04 ±0.97 | 80.89 ±0.51 | 89.67 ±0.88 | 80.75 ±2.64 | 77.93 ±4.37 | 35.94 ±1.29 |
| PSL-GCN | **89.09±1.14** | **88.75±1.24** | **81.04±0.63** | **91.48±0.81** | **81.45±2.43** | **78.13±4.32** | **36.74±1.43** |
| APPNP | 88.23±1.05 | 88.17±0.38 | 80.61±0.94 | 90.48±0.57 | 90.89±1.53 | 91.84±2.12 | 39.59±0.83 |
| GSL-APPNP | 89.88±0.95 | 88.46±1.38 | 81.52±1.16 | 90.89±0.93 | 91.19±1.47 | 91.93±2.23 | 39.81±0.84 |
| PSL-APPNP | **90.91±1.14** | **88.88±1.42** | **81.92±1.36** | **91.82±0.96** | **91.53±1.49** | **92.07±1.94** | **40.67±0.91** |
| Graph-Heat | 87.34±1.41 | 87.52±0.89 | 78.82±1.26 | 89.97±0.82 | 80.06±0.94 | 76.84±0.85 | 35.74±1.31 |
| GSL-Heat | 87.71±1.22 | 89.14±0.89 | 79.23±1.02 | 90.34±1.31 | 80.79±1.9 | 80.87±0.70 | 36.36±1.87 |
| PSL-Heat | **88.51±1.38** | **89.21±1.57** | **80.02±1.31** | **90.87±0.91** | **81.49±1.10** | **80.94±1.05** | **36.93±1.45** |
| BernNet | 88.48±1.14 | 88.73±1.42 | 80.08±1.07 | 94.31±0.86 | 94.14±1.67 | 92.16±1.77 | 41.82±0.99 |
| GSL-BernNet | 89.96±1.11 | 89.84±1.50 | 80.77±0.81 | 94.67±1.10 | 94.23±1.76 | 92.68±1.07 | 41.88±1.44 |
| PSL-BernNet | **90.01±0.93** | **90.13±1.44** | **80.86±1.15** | **94.79±1.03** | **94.54±1.80** | **94.01±0.93** | **41.93±1.30** |
| GPRGNN | 88.54±0.82 | 88.52±0.46 | 80.09±1.03 | 93.87±0.34 | 92.97±1.41 | 91.32±2.02 | 39.92±0.65 |
| GSL-GPR | 89.82±1.23 | 89.04±1.01 | 80.56±0.93 | 94.12±1.25 | 93.47±1.38 | 91.87±1.72 | 40.64±0.82 |
| PSL-GPR | **90.13±0.92** | **89.82±1.89** | **81.11±0.76** | **94.87±0.96** | **94.74±1.65** | **92.44±1.85** | **41.17±0.96** |
| Jacobi | 88.96±0.68 | 89.67±0.82 | 80.73±0.88 | 95.52±0.33 | 93.45±2.03 | 92.94±2.38 | 41.16±0.70 |
| GSL-Jacobi | 89.43±0.94 | 89.92±1.33 | 81.24±1.16 | 95.55±0.36 | 93.84±1.97 | 93.45±1.71 | 41.21±0.45 |
| PSL-Jacobi | **90.73±1.34** | **90.42±1.13** | **81.54±1.12** | **95.69±0.52** | **94.20±1.75** | **93.91±1.96** | **41.94±0.75** |

It can be observed that PSL outperforms GSL, further validating the effectiveness of the perturbed sheaf Laplacian.

