# OpenReview forum: "Enhancing Spectral GNNs: From Topology and Perturbation Perspectives"
_ICML.cc/2025/Conference — ICML 2025 poster_

### Official Review · Reviewer_wYZy · 2025-03-05

**Overall Recommendation:** 3

**Summary:**

This paper proposes a higher-dimensional sheaf Laplacian matrix based on perturbation theory and the theory of cellular sheaves. The perturbation is controlled from the block form of the normalized graph Laplacian matrix, and can contain more distinct eigenvalues. The paper provides theoretical analyses on the expressiveness of spectral GNNs and perturbation bounds for the eigenvalues. Node classification experiments demonstrate the efficacy of the proposed Laplacian.

**Claims And Evidence:**

The claims seem clear and convincing.

**Essential References Not Discussed:**

N/A

**Experimental Designs Or Analyses:**

1. Table 2 shares some information with Table 1, but with different performance values. Is this due to execution variation? Could you merge the table instead?
2. Why do you not compare your proposed PSL with the previous Sheaf Laplacian based GNNs mentioned in Sec. 7.2?
3. The ablation study part does not discuss model complexity (which may offer another angle of why PSL-GNN performs better, but may not be an advantage). I think adding runtime comparison may be useful.

**Methods And Evaluation Criteria:**

The proposed method seems valid, but may have much larger complexity. The paper mentions the construction complexity for the Laplacian, but not the GNN complexity induced by using a more complex Laplacian.

**Other Comments Or Suggestions:**

It may be better if you first write out Proposition 5.1 to motivate the need of perturbation, then introduce perturbations.

**Other Strengths And Weaknesses:**

The paper is generally well-written and flows well.

**Questions For Authors:**

1. What is the relationship between Tables 1 and 2?
2. Why do you not compare your proposed PSL with the previous Sheaf Laplacian based GNNs mentioned in Sec. 7.2?
3. What is the model complexity in terms of GNN computation, not just construction of the Laplacian?

**Relation To Broader Scientific Literature:**

The contribution is related to the other Sheaf Neural Networks on graphs but seems to have some advantages (not numerically compared though).

**Theoretical Claims:**

The theorems and propositions seem correct.

---

> ### Author Rebuttal · Authors · 2025-03-27
>
> Thank you for your valuable suggestion.
>
> **Response to Q1**
>
> We apologize for any confusion caused. To clarify: Table 1 demonstrates the performance gains achieved by integrating PSL into various models, while Table 2 highlights the comparison between PSL, GSL, and the conventional normalized graph Laplacian.  Table 2 shares some information with Table 1, but with different performance values. It is because we retrained all the models separately under the same experimental environment. Therefore, although the experimental results differ slightly, these variations fall within the normal fluctuation range. As suggested by other reviewers, we reconducted comprehensive experiments by adding four new datasets (10 in total). All baselines use their optimal parameter settings as reported in their original papers. The new Table 1 is available at <https://anonymous.4open.science/r/exp1-DF26/exp1.pdf>, and the new  Table 2  is available at <https://anonymous.4open.science/r/exp1-DF26/exp2.pdf>, ensuring the robust validation of the proposed method.
>
> **Response to Q2**
>
> Next, we will explain the reason for not comparing the previous Sheaf Laplacian-based GNNs in Sec. 7.2.
>
> SheafNN by Hansen and Gebhart was designed under the settings with a known sheaf structure, but real-world datasets typically lack this information.
>
> Bodnar et al. adapted the sheaf Laplacian into a learnable form that is suitable for graph learning, making it applicable for graph learning. Our GSL indeed follows Bodnar et al.'s approach, so we have already compared our approach with their proposed Sheaf Laplacian-based GNN.
>
> The hypergraph sheaf Laplacian proposed by Duta et al. is explicitly designed for hypergraphs, making it inapplicable to our benchmark datasets.
>
> Both Conn-NSD by Barbero et al. and D-TNN by Battiloro et al. impose restrictive assumption on high node degrees, which are not satisfied by most benchmark graph datasets.
>
>
> **Response to Q3**
>
> **Notation:**
> - $n$: Number of nodes
> - $f_1$, $f_3$, $f_4$: Feature dimensions
> - $d$: Stalk dimension
> - $c$: Number of classes
> - $\text{nnz}_{\mathcal{B}}$: Number of nonzeros in the edge incidence matrix (approximately $2E$, where $|E|$ is the number of edges)
> - $r$: Average number of nonzeros per row in the PSL matrix
> - $f_{last}$: Feature dimension of the last layer
>
> For PSL-GCN, we generate a new PSL every 10 epochs; therefore, for efficiency analysis per epoch, we consider the following two cases. In the first case we include PSL construction time. The preprocessing stage has a complexity of $O\big(n \cdot f_1 \cdot d \cdot f_3\big)$.
>
> The PSL construction has a complexity of $O\big(\text{nnz}_{\mathcal{B}}^2\big)$.
>
> The forward propagation for 2 layers has a complexity of $O\big(2 \cdot nd \cdot f_3 \cdot (r + f_4)\big)$ (with $r$ representing the average number of nonzeros per row in the PSL matrix).
> The final output transformation has a complexity of $O\big(n \cdot d \cdot f_{last} \cdot c\big)$.
>
> Thus, the total complexity is
> $$O\Big(\text{nnz}_{\mathcal{B}}^2 + n \cdot f_1 \cdot d \cdot f_3 + 2 \cdot nd \cdot f_3 (r + f_4) + n \cdot d \cdot f_{last} \cdot c\Big).$$
>
> In the second case, we exclude PSL construction time, and the complexity per epoch is
> $$O\Big(n \cdot f_1 \cdot d \cdot f_3 + 2 \cdot nd \cdot f_3 (r + f_4) + n \cdot d \cdot f_{last} \cdot c\Big).$$
>
> In order to empirically validate the efficiency of our method, we compare it with the representative work [4] *Neural Sheaf Diffusion: A Topological Perspective on Heterophily and Oversmoothing in GNNs*. We report the average epoch runtime (ms) and the total runtime per fold (s) in the table below:
>
> | Datasets   | Cora       | Pubmed      | Citeseer   | Photo      | Texas    | Cornell  | Actor      |
> |------------|------------|-------------|------------|------------|----------|----------|------------|
> | Diag-NSD   | 26.4/5.9   | 124.3/34.7  | 25.2/5.8   | 90.6/17.4  | 8.3/1.6  | 8.1/1.5  | 98.4/21.5  |
> | O(d)-NSD   | 57/12.3    | 204.5/66.3  | 64.1/13.6  | 143.5/26.3 | 26.8/6.7 | 27.4/7.2 | 128.2/31.9 |
> | Gen-NSD    | 85.3/17.6  | 231.1/74.3  | 92.4/20.6  | 166.7/34.6 | 34.3/14.4| 31.4/14.2| 177.6/45.2 |
> | PSL-GCN    | 16/4.6     | 83.7/17.4   | 18.3/5.2   | 65.3/15.8  | 6.3/1.1  | 6.4/1.1  | 72.7/17.3  |

---

### Official Review · Reviewer_nURk · 2025-03-06

**Overall Recommendation:** 4

**Summary:**

This paper claims that the presence of repeated eigenvalues limits the expressive power of spectral GNNs. To address this issue, this paper proposes perturbed sheaf Laplacian, which achieves optimal model performance due to its more distinct eigenvalues.

**Claims And Evidence:**

The occurrence of repeated eigenvalues does indeed limit the expression ability of spectral GNN, as evidenced by many previous literature. However, this paper claims that eigenvalue correction [1] damages the topological information of the original graph, but does not provide any analysis or basis.

[1]Lu, K., Yu, Y., Fei, H., Li, X., Yang, Z., Guo, Z., Liang, M., Yin, M., and Chua, T.-S. Improving expressive power of spectral graph neural networks with eigenvalue correction. In Proceedings of the AAAI Conference on Artificial Intelligence, volume 38, pp. 14158–14166, 2024.

**Essential References Not Discussed:**

The eigenvalue correction method [1] can also solve the problem of duplicate eigenvalues, but this paper did not compare it with it.
[1]Lu, K., Yu, Y., Fei, H., Li, X., Yang, Z., Guo, Z., Liang, M., Yin, M., and Chua, T.-S. Improving expressive power of spectral graph neural networks with eigenvalue correction. In Proceedings of the AAAI Conference on Artificial Intelligence, volume 38, pp. 14158–14166, 2024.

**Experimental Designs Or Analyses:**

1.According to Table 2, the performance of PSL-GNN is comparable to that of GSL-GNN. Does this indicate that PSL-GNN is sufficient to achieve optimal performance without the need for GSL-GNN?
2. Obviously, this paper improves based on eigenvalue correction, but does not compare it with eigenvalue correction.
3.Although this paper provides a complexity analysis, the actual runtime is helpful for readers to understand the scalability of the proposed method.

**Methods And Evaluation Criteria:**

Yes, the proposed method helps alleviate the problem of limited expressiveness caused by repeated eigenvalues.

**Other Comments Or Suggestions:**

No

**Other Strengths And Weaknesses:**

Compared to eigenvalue correction, this paper does not require expensive eigenvalue decomposition.

**Questions For Authors:**

1.What are the advantages of this paper compared to eigenvalue correction? Although the paper mentions that eigenvalue correction damages the topological information of the original graph, it does not provide a detailed explanation. And this paper did not compare it with eigenvalue correction.
2.What is the training efficiency of the proposed method? Obviously, this paper introduces more intensive matrix operations, so providing actual training time is beneficial.

**Relation To Broader Scientific Literature:**

Previous literature has shown that repeated eigenvalues can limit the expressive power of spectral neural networks and hinder model performance. This paper adopts a novel approach (perturbed sheaf Laplacian) to solve this problem.

**Theoretical Claims:**

No

---

> ### Author Rebuttal · Authors · 2025-03-27
>
> Thank you for your valuable suggestion.
>
> **Response to Comment 1 in Experimental Designs or Analyses**
>
> We reconducted the experiments by adding four new datasets (10 in total). All baseline algorithms use the best parameter settings in their original paper. Additionally, we also tuned the parameters in our proposed method.
>
>
> Following are the partial experimental results,  and the complete experimental results are available at <https://anonymous.4open.science/r/exp1-DF26/exp2.pdf>.
>
> | Datasets    | Cora         | Pubmed       | Citeseer     | Photo        | Texas        | Cornell      | Actor        |
> |-------------|--------------|--------------|--------------|--------------|--------------|--------------|--------------|
> | Jacobi      | 88.96±0.68   | 89.67±0.82   | 80.73±0.88   | 95.52±0.33   | 93.45±2.03   | 92.94±2.38   | 41.16±0.70   |
> | GSL-Jacobi  | 89.43±0.94   | 89.92±1.33   | 81.24±1.16   | 95.55±0.36   | 93.84±1.97   | 93.45±1.71   | 41.21±0.45   |
> | PSL-Jacobi  | **90.73±1.34** | **90.42±1.13** | **81.54±1.12** | **95.69±0.52** | **94.20±1.75** | **93.91±1.96** | **41.94±0.75** |
>
> The results show that PSL performs better than GSL. Nonetheless, we believe that GSL merits further investigation, as it can theoretically learn a new operator with fewer or even no repeated eigenvalues. In this paper, PSL is a special case of GSL, which uses perturbation to achieve more distinct eigenvalues, effectively alleviating the problem of repeat eigenvalues. However, our current perturbation method is not refined enough to completely eliminate repeated eigenvalues—a task that would require a deeper understanding of the sheaf Laplacian's properties, a challenging problem we aim to address in future, as mentioned in the conclusion of our paper.
>
>
> **Response to Comment 2 in Experimental Designs or Analyses**
>
> We included the eigenvalue correction method in the baselines and compared vanilla GNN, EC-GNN, and PSL-GNN in the new experimental settings.The results can be found at <https://anonymous.4open.science/r/exp1-DF26/exp1.pdf>. The experiments show that our method improves all baselines including EC-GNN.
>
> **Response to Comments 3 in Experimental Designs or Analyses**
>
> Please refer to the following Response to Q2 in Questions For Authors.
>
>
> **Response to Q1 in Questions For Authors**
>
> Compared with the eigenvalue correction method, our proposed PSL does not compromise the original topological information encoded in the normalized Laplacian matrix. Here, we briefly explain why the eigenvalue correction method will lead to this issue. The spectral gap of the normalized Laplacian often measures the quality of graph connectivity. The eigenvalue correction method, however, reduces the spectral gap of the new operator H (i.e., u₁ = βλ₁ + (1-β)v₁< λ₁, when v₁ < λ₁), which in turn affects information propagation (please see [3]: Spectral Graph Pruning Against Over-Squashing and Over-Smoothing for details). We also experimentally validated this effect:
>
> |             | Cora  | Pubmed | Citeseer | Photo | Texas | Cornell | Actor |
> |-------------|-------|--------|----------|-------|-------|---------|-------|
> | **n**     | 2708  | 19717  | 3327     | 7650  | 183   | 183     | 7600  |
> | **v₁**    | 7e-4  | 1e-4   | 6e-4     | 2e-4  | 1e-2  | 1e-2    | 2e-4  |
> | **λ₁**    | 4e-3  | 1e-2   | 1e-3     | 1e-3  | 5e-2  | 7e-2    | 3e-2  |
> | **λ̂₁**  | 5e-3  | 1e-2   | 2e-3     | 1e-3  | 5e-2  | 7e-2    | 3e-2  |
>
> Here,  $\hat{\lambda}_1$ denotes the spectral gap of PSL.  The comparison between EC-GNN and PSL-GNN is available at <https://anonymous.4open.science/r/exp1-DF26/exp1.pdf>. The results show that PSL-GNN outperforms EC-GNN, demonstrating the superiority of our work.
>
> **Response to Q2 in Questions For Authors**
>
> We compared the efficiency of PSL-GNN with a representative work ([4] *Neural Sheaf Diffusion: A Topological Perspective on Heterophily and Oversmoothing in GNNs*). Following the experimental setup in [4], we compared PSL-GCN with the three GCN-based sheaf models proposed in [4] and reported the average epoch runtime (ms) and the total runtime per fold (s). The results are shown in the table below:
>
> | Datasets   | Cora       | Pubmed      | Citeseer   | Photo      | Texas    | Cornell  | Actor      |
> |------------|------------|-------------|------------|------------|----------|----------|------------|
> | Diag-NSD   | 26.4/5.9   | 124.3/34.7  | 25.2/5.8   | 90.6/17.4  | 8.3/1.6  | 8.1/1.5  | 98.4/21.5  |
> | O(d)-NSD   | 57/12.3    | 204.5/66.3  | 64.1/13.6  | 143.5/26.3 | 26.8/6.7 | 27.4/7.2 | 128.2/31.9 |
> | Gen-NSD    | 85.3/17.6  | 231.1/74.3  | 92.4/20.6  | 166.7/34.6 | 34.3/14.4| 31.4/14.2| 177.6/45.2 |
> | PSL-GCN    | 16/4.6     | 83.7/17.4   | 18.3/5.2   | 65.3/15.8  | 6.3/1.1  | 6.4/1.1  | 72.7/17.3  |
>
> As we can see our approach is significantly more efficient than the other sheaf models.

---

### Official Review · Reviewer_oqeP · 2025-03-13

**Overall Recommendation:** 3

**Summary:**

This paper aims to solve the repeated eigenvalues of graph Laplacian by proposing a novel perturbed sheaf Laplacian (PSL). The authors claim that PSL can increase the number of distinct eigenvalues and improve the expressive power of spectral GNNs. Experiments on the node classification task validate the effectiveness of PSL on different spectral GNNs.

**Claims And Evidence:**

Some of the claims have been confirmed, but others remain unconvincing.

- C1: **It also compromises the original topological information encoded in the normalized Laplacian matrix.** It is unclear why previous methods fail to use the topological information. This paper claims that "PSL can retain the topological information of the normalized Laplacian matrix", implying that it has the same topological information of the original graph Laplacian. As a result, I think previous methods can also leverage the topological information.

- C2: **Comparion between PSL and GSL**. The proposed PSL is a perturbed version of the general sheaf Laplacian (GSL). However, in the ablation studies, the performance of PSL does not outperform GSL by a large margin. Therefore, the audience may doubt the effectiveness of PSL.

**Essential References Not Discussed:**

This paper only applies PSL to the polynomial GNNs, which is a part of spectral GNNs. It would be better if the authors can try different architectures of spectral GNNs, such as Specformer [3].

[3] Specformer: Spectral Graph Neural Networks Meet Transformers. ICLR 2023.

**Experimental Designs Or Analyses:**

This paper should add graph-level experiments. See Methods And Evaluation Criteria.

**Methods And Evaluation Criteria:**

1. This paper only conducts experiments on the node classificiation datasets, which is unconvincing. In practice, we often use graph-level tasks, such as structure counting, to evaluate the expressive power of GNNs. See [1] for more details.

2. This paper does not report the performance of Lu et al., 2024 [2], making the results less convincing.

[1] Graph as Point Set. ICML 2024.

[2] Improving expressive power of spectral graph neural networks with eigenvalue correction. AAAI 2024.

**Other Comments Or Suggestions:**

See the above comments.

**Other Strengths And Weaknesses:**

See the above comments.

**Questions For Authors:**

See the above comments.

**Relation To Broader Scientific Literature:**

This paper introduces PSL, which is a model-agnostic method to improve the expressive power of spectral GNNs. PSL consistently improves the performance of polynomial GNNs, which may benefit some downstream applications.

**Theoretical Claims:**

Roughly checking but not confident.

---

> ### Author Rebuttal · Authors · 2025-03-26
>
> Thank you for your valuable suggestion.
>
> **Response to C1**
>
> We apologize for the confusion.
> Briefly, the Laplacian spectrum measures graph connectivity. Cheeger's inequality $2h_G > \lambda_1 > \frac{h_G^2}{2}$ shows that a larger spectral gap $\lambda_1$ implies better connectivity (More details can be found in: Spectral Graph Pruning Against Over-Squashing and Over-Smoothing). The Cheeger constant $h_G$ quantifies the tightest connectivity bottleneck in a graph. In Lu et al.'s method, the new operator's ($H$) eigenvalues are defined as $u_i = \beta \lambda_i + (1-\beta)v_i$, where $v_i = 2i/n - 2$. When $v_1 < \lambda_1$, it follows that $u_1 < \lambda_1$ (because $u_1 - \lambda_1 = (1 - \beta)(v_1 - \lambda_1)$ with $\beta < 1$), which reduces the spectral gap and impairs information flow, especially in large graphs, where $v_1 = 2/n - 2$ is very, very small. However, our approach perturbs the normalized Laplacian in a way that maintains subtle eigenvalue differences regardless of graph size, thereby preserving the topological information.
>
> Below, we verify our argument above by presenting the spectral gaps' orders of magnitude for the normalized graph Laplacian and PSL across various datasets. As shown in the following table, for each dataset, $v_1 < \lambda_1$, $u_1 < \lambda_1$ , and $\hat{\lambda}_1 \leq \lambda_1$, which indicates that Lu et al.'s approach indeed reduces the spectral gap, thus compromising the original topological information encoded in the normalized Laplacian matrix.
>
> |   | Cora  | Pubmed | Citeseer | Photo | Texas | Cornell | Actor |
> |-------------|-------|--------|----------|-------|-------|---------|-------|
> | **n**     | 2708  | 19717  | 3327     | 7650  | 183   | 183     | 7600  |
> | **v₁**    | 7e-4  | 1e-4   | 6e-4     | 2e-4  | 1e-2  | 1e-2    | 2e-4  |
> | **λ₁**    | 4e-3  | 1e-2   | 1e-3     | 1e-3  | 5e-2  | 7e-2    | 3e-2  |
> | **λ̂₁**  | 5e-3  | 1e-2   | 2e-3     | 1e-3  | 5e-2  | 7e-2    | 3e-2  |
>
> **Response to C2**
>
> According to Reviewer QU1C's suggestion, we reconducted comprehensive experiments by adding four new datasets (10 in total). All baseline algorithms use the best parameter settings in their original papers. The results demonstrate PSL achieves better performance than GSL, which are available at <https://anonymous.4open.science/r/exp1-DF26/exp2.pdf>.
>
> **Response to Q1 in Methods And Evaluation Criteria**
>
> It's insightful to point out the relevance of graph-level tasks in [1]. The primary focus of our paper was to address a specific known limitation inherent to spectral GNNs: the performance degradation and limited expressiveness caused by repeated eigenvalues in the standard graph Laplacian. We chose node classification to demonstrate direct evidence that our proposed PSL method successfully overcomes the targeted limitation in a practical application for spectral GNNs.
>
> Standard spectral GNN approaches face inherent challenges when applied across datasets containing graphs of varying sizes and structures, which are common in graph-level tasks. It's difficult to directly learn a single spectral filter that works across different $\Lambda$ and $U$ matrices from different graphs.
>
> Rigorously evaluating PSL's impact on graph-level tasks might involve developing novel ways to combine PSL-based spectral features with appropriate graph pooling mechanisms. While valuable, we considered this beyond the primary scope of introducing and validating the core PSL concept.
>
> **Response to Q2 in Methods And Evaluation Criteria**
>
> We included Lu et al.'s method in the baselines. The results in the following table show our method outperforms Lu et al.'s method. The full results are available at <https://anonymous.4open.science/r/exp1-DF26/exp1.pdf>.
> | Datasets    | Cora          | Citeseer      | PubMed        | Computers     | Photo         | Chameleon      | Actor         | Squirrel      | Texas         | Cornell       |
> |-------------|---------------|---------------|---------------|---------------|---------------|----------------|---------------|---------------|---------------|---------------|
> | Jacobi      | 88.96±0.68    | 80.73±0.88    | 89.67±0.82    | 90.42±0.31    | 95.52±0.33    | 74.23±1.45     | 41.16±0.70    | 57.38±1.24    | 93.45±2.03    | 92.94±2.38    |
> | EC-Jacobi (Lu et al.'s method)   | 89.06±0.67    | 81.28±0.96    | 89.87±0.42    | 90.33±0.28    | 95.54±0.36   | 75.64±1.51     | 41.01±0.74    | 59.87±0.91    | 93.48±1.49    | 93.29±2.33    |
> | PSL-Jacobi (Our method) | **90.73±1.34** | **81.54±1.12** | **90.42±1.13** | **90.83±0.61** | **95.69±0.52** | **75.87±1.44** | **41.94±0.75** | **61.47±0.98** | **94.20±1.75** | **93.91±1.96** |
>
> **Response to comments in Essential References Not Discussed**
>
> Specformer requires eigenvalue encoding. If PSL is integrated into Specformer, we have to encode the learned PSL, which implies the encoding process would involve training a PSL-GNN. The computation cost would be substantially increased.

---

> > ### Comment · Reviewer_oqeP · 2025-04-03
> >
> > I have read the authors' rebuttal and checked other reviewers' comments. Although this paper does not provide experiments on graph-level tasks, it still brings a good idea and new content to GNNs. I raise my score to weak accept.

---

### Official Review · Reviewer_QU1C · 2025-03-21

**Overall Recommendation:** 3

**Summary:**

This paper presents a novel solution to the repeated eigenvalue problem in Spectral GNNs.

Through the formal definition of cellular sheaf on graphs,
the paper formally introduces the definition of cellular sheaf,
which essentially specifies that when a signal with dimension $d$ propagates from node $i$ along edge $(i,j)$ to another node $j$,
it is not directly added but first undergoes a linear transformation, which can be denoted as $Q_{ij}$.
Each $Q_{ij}$ contains $d\times 1$ learnable parameters.


Assuming this linear transformation is Identity, the operating matrix only changes from L to $L \otimes I_d$,
with the same number of distinct eigenvalues
(this operation can be viewed as expanding the dimension of $L$).
Now, since this linear transformation is not Identity, it actually brings a slight perturbation,
which increases the number of distinct eigenvalues.
The paper combines Weyl's theorem to show that when this perturbation is small enough, it guarantees an increase in distinct eigenvalues.
And through experiments, it verifies that indeed more eigenvalues are obtained.

The paper compares with state-of-the-art polynomial-filter-based GNNs with almost no hyperparameter tuning,
and shows that the Sheaf Laplacian method brings a slight advantage.

**Claims And Evidence:**

- The part about introducing Sheaf Laplacian and the perturbation method to bring more distinct eigenvalues is very interesting and clear and convincing.

- It undoubtedly makes a contribution to the multiple eigenvalue problem.


- However, I'm not entirely convinced that the experiments sufficiently support the claim that ``solving the multiple eigenvalue problem can improve node classification performance''. Please see section (5. Experimental Designs Or Analyses) in the Review table.

**Essential References Not Discussed:**

No.

**Experimental Designs Or Analyses:**

The paper's experiments are the common node classification tasks in the Spectral GNN series of work,
including both homophilic and heterophilic graph datasets.
However, I'm concerned that the way authors conducted the experiments seems too rough.

Spectral GNNs, including GCN, are sensitive to hyperparameters (of course, the tendency to over-tune hyperparameters is a bad problem),
but this paper, according to Appendix E.4., seems to have done no hyperparameter tuning at all.
Therefore, the experimental data reported in the paper is significantly **lower** than elsewhere.
Meanwhile, the method presented in the paper only brings quite **slight** improvements,
and the number of datasets compared is also relatively small.

So, although I appreciate the ideas in this work,
I'm skeptical about whether solving the repeated eigenvalue problem can truly bring improvements to downstream tasks, and think the experiments are insufficient.

**Methods And Evaluation Criteria:**

Yes.

**Other Comments Or Suggestions:**

None.

**Other Strengths And Weaknesses:**

Please check the former review form(2. Claims And Evidence)

**Questions For Authors:**

I mainly care the issues raised in section (5. Experimental Designs Or Analyses).

1. I suggest the authors report the difference in response values of $g(\lambda)$ before and after perturbation.

2. I suggest the authors use Optuna for light hyperparameter tuning to confirm whether the method indeed has advantages (empirically).

3. Could the authors explain more on the correspondence between $\nu$ and $\|P\|$ in Thm.4.1? If there is theoretical justification, that would be great. If not, empirical evidence would also be helpful to understand this relationship.

**Relation To Broader Scientific Literature:**

This paper mainly responds to the discussion about repeated eigenvalues' impact on spectral GNNs expressiveness in Wang & Zhang (2022).

**Theoretical Claims:**

I roughly checked Theorem 4.1 and Prop 5.2 - Corollary 5.4.

---

> ### Author Rebuttal · Authors · 2025-03-26
>
> Thank you for your valuable suggestion.
>
> **Response to the raised issues in Experimental Designs or Analyses**
>
> We reconducted the experiments by adding four new datasets (10 total) and using a full-supervised split (60%/20%/20%), which follows Wang & Zhang’s (*How Powerful are Spectral Graph Neural Networks*). It is worth noting that our previous experiments use the split (48\%/32\%/20\%), which is only recommended by Bondar et al. in *Neural Sheaf Diffusion: A Topological Perspective on Heterophily and Oversmoothing in GNNs*.
>
> The partial experimental results are as follows, in which each baseline algorithm uses the best parameter settings in their original paper. We also tuned the parameters in our proposed method. It shows that PSL-GNN still yields improvements over all baselines and performs better than EC-GNN.
>
> The complete experimental results are available at <https://anonymous.4open.science/r/exp1-DF26/exp1.pdf>.
>
>
> | Datasets    | Cora          | Citeseer      | PubMed        | Computers     | Photo         | Chameleon      | Actor         | Squirrel      | Texas         | Cornell       |
> |-------------|---------------|---------------|---------------|---------------|---------------|----------------|---------------|---------------|---------------|---------------|
> | GPRGNN      | 88.54±0.82    | 80.09±1.03    | 88.52±0.46    | 87.01±0.74    | 93.87±0.34    | 67.14±1.10     | 39.92±0.65    | 50.08±1.95    | 92.97±1.41    | 91.32±2.02    |
> | EC-GPR      | 89.41±0.69    | 80.66±1.01    | 89.64±0.53    | 89.91±0.68    | 94.76±1.02    | 74.24±1.06     | 40.42±0.77    | 62.48±2.03    | 92.27±1.92    | 90.79±2.22    |
> | PSL-GPR     | **90.13±0.92** | **81.11±0.76** | **89.82±1.89** | **89.93±1.70** | **94.87±0.96** | **74.78±1.34** | **41.17±0.96** | **63.65±0.87** | **94.74±1.65** | **92.44±1.85** |
> | Jacobi      | 88.96±0.68    | 80.73±0.88    | 89.67±0.82    | 90.42±0.31    | 95.52±0.33    | 74.23±1.45     | 41.16±0.70    | 57.38±1.24    | 93.45±2.03    | 92.94±2.38    |
> | EC-Jacobi   | 89.06±0.67    | 81.28±0.96    | 89.87±0.42    | 90.33±0.28    | 95.54±0.36   | 75.64±1.51     | 41.01±0.74    | 59.87±0.91    | 93.48±1.49    | 93.29±2.33    |
> | PSL-Jacobi  | **90.73±1.34** | **81.54±1.12** | **90.42±1.13** | **90.83±0.61** | **95.69±0.52** | **75.87±1.44** | **41.94±0.75** | **61.47±0.98** | **94.20±1.75** | **93.91±1.96** |
>
> **Response to Question 1**
>
> We design a metric $S(k,m)$ to quantify the difference in response values of $g(\lambda)$ before and after perturbation. Assuming the filtering coefficients for frequency components $U^T X_i W$ before and after perturbation are $k$ and $m$, respectively, the similarity metric is defined as:
> $$
> S(k,m)=
> \begin{cases}
> 1, & \text{if } k = m = 0, \\
> 1 - 2\frac{|k-m|}{|k|+|m|}, & \text{otherwise}.
> \end{cases}
> $$
> Specifically, $S(k,m)=1$ indicates identical coefficients ($k=m$), including when both are zero. We measured the response difference for PSL-GCN and PSL-GPR in the Cora dataset. The figure showing the results is at  <https://anonymous.4open.science/r/exp1-DF26/response.pdf>. Only about 1/10 of all nodes exhibit no change ($S=1$) in their filtering coefficients before and after perturbation, which aligns with the results reported in Table 3 of our paper.
>
> **Response to Question 2**
>
> Please refer to Response to the raised issues in Experimental Designs or Analyses.
>
> **Response to Question 3**
>
> We guess you mean $\phi$. Recall that in Theorem 4.1, $\phi$ is defined as the minimum eigenvalue gap of the normalized Laplacian matrix before perturbation. When a perturbation matrix $P$ is applied, Weyl’s inequality tells us that the eigenvalue change is bounded by its spectral norm, $||P||_2$. So if $||P||_2 < \phi$, then the eigenvalue variation intervals do not overlap, resulting in no eigenvalue multiplicity. Otherwise, the new eigenvalues might coincide.
>
> While Theorem 4.1 describes a strict condition for ensuring fully distinct eigenvalues, in practice, it is not required to adhere to this condition strictly. We only need $||P||_2$ to be sufficiently small (but not too small. $||P||_2$ approaching zero would alleviate the perturbation effects.).
> To address this, we introduce the learnable perturbation restriction maps to achieve a controllable and appropriate perturbation matrix $P$. Empirically, this approach maintains a sufficiently small spectral norm such that more eigenvalues split without generating identical new eigenvalues.

---

> > ### Comment · Reviewer_QU1C · 2025-04-06
> >
> > Thank you for the supplementary experimental details.
> >
> > Regarding **Q3**, I would like to ask whether in your experiments, you actually demonstrated the $\phi$ in the Theorem through $\nu$?

---

> > > ### Author Response · Authors · 2025-04-07
> > >
> > > Thank you so much for prompting this clarification. Our paper uses the symbols $\phi$ and $\eta$, but just to confirm, $\nu$ does not appear in our work. That might reflect a misunderstanding in our previous response. Thinking about parameters that might be closely related to your query for this final response, we considered the potential relevance of $\eta$. We conducted an empirical study on $\eta$'s impact on eigenvalue perturbations and performance, and the findings are reported in Appendix F.2 and F.4 in our paper. We hope this addresses your question more fully!

---

### Decision · Program_Chairs · 2025-05-01

**Decision:**

Accept (poster)

**Comment:**

The reviewers unanimously recommend acceptance of the paper with varying degrees of strength. I agree with their assessment and am happy to recommend acceptance of this paper.

The reviewers have highlighted a need for more benchmark datasets, raised questions about the hyperparameter tuning, noticed a lack of comparison to an existing eigenvalue correction method and discussed the computational complexity of the proposed method. In my view, these topics were all convincingly resolved by the reviewers in their rebuttal. The only unaddressed point is a lack of evaluation on graph-level learning tasks. However, the authors argue that graph-level learning tasks present a challenge for many spectral GNNs. So, while this comment remains open. I do not think that it justifies a rejection decision, as it indeed is a problem that is not specific to the submitted work. I want to nonetheless encourage the authors to mention this limitation in their manuscript.